EMBO
Molecular Medicine

# Impaired telomere integrity and rRNA biogenesis in PARN-deficient patients and knock-out models

Maname Benyelles[1,2], Harikleia Episkopou[3], Marie-Françoise O'Donohue[4], Laëtitia Kermasson[1,2], Pierre Frange[5,6], Florian Poulain[3], Fatma Burcu Belen[7], Meltem Polat[8], Christine Bole-Feysot[9,10], Francina Langa-Vives[11], Pierre-Emmanuel Gleizes[4], Jean-Pierre de Villartay[1,2], Isabelle Callebaut[12], Anabelle Decottignies[3,*,†] (iD) & Patrick Revy[1,2,**,†] (iD)

## Abstract

PARN, poly(A)-specific ribonuclease, regulates the turnover of mRNAs and the maturation and stabilization of the h*TR* RNA component of telomerase. Biallelic *PARN* mutations were associated with Høyeraal–Hreidarsson (HH) syndrome, a rare telomere biology disorder that, because of its severity, is likely not exclusively due to h*TR* down-regulation. Whether PARN deficiency was affecting the expression of telomere-related genes was still unclear. Using cells from two unrelated HH individuals carrying novel *PARN* mutations and a human PARN knock-out (KO) cell line with inducible *PARN* complementation, we found that PARN deficiency affects both telomere length and stability and down-regulates the expression of *TRF1*, *TRF2*, *TPP1*, *RAP1*, and *POT1* shelterin transcripts. Down-regulation of dyskerin-encoding *DKC1* mRNA was also observed and found to result from p53 activation in PARN-deficient cells. We further showed that PARN deficiency compromises ribosomal RNA biogenesis in patients' fibroblasts and cells from heterozygous *Parn* KO mice. Homozygous *Parn* KO however resulted in early embryonic lethality that was not overcome by *p53* KO. Our results refine our knowledge on the pleiotropic cellular consequences of PARN deficiency.

**Keywords** Høyeraal–Hreidarsson syndrome; p53; PARN; rRNA; shelterin
**Subject Categories** Chromatin, Epigenetics, Genomics & Functional Genomics; Genetics, Gene Therapy & Genetic Disease

## Introduction

Telomeres represent ribonucleoprotein complexes at the end of linear chromosomes and are composed of *TTAGGG* repeats in mammals. Because conventional DNA polymerases are unable to fully replicate chromosome ends, telomeres progressively shorten over successive cell divisions resulting in the production of short telomeres that induce an irreversible cell cycle arrest, known as replicative senescence (Blackburn *et al*, 2006). Telomere shortening can be overcome by hTERT, the reverse transcriptase component from the telomerase complex that also comprises dyskerin, the accessory factors NHP2, GAR1, NOP10, and h*TR/TERC* used as RNA template to produce telomeric sequences. Human telomerase is only active in germ cells, most cancer cells, and some stem or activated cells. Telomeric sequences warrant the binding of TRF1 and TRF2 that interact with POT1, TPP1, RAP1, and TIN2 to form the shelterin complex that protects telomeres from degradation and fusion and regulates telomerase recruitment and activity (de Lange, 2018).

In humans, innate defects resulting in excessive shortening or impaired protection of telomeres cause a large spectrum of diseases including pulmonary fibrosis, aplastic anemia, dyskeratosis congenita (DC), and Høyeraal–Hreidarsson (HH) or Revesz syndromes

1   Laboratory of Genome Dynamics in the Immune System, INSERM, UMR 1163, Paris, France
2   Laboratoire labellisé Ligue, Imagine Institute, Paris Descartes–Sorbonne Paris Cite University, Paris, France
3   de Duve Institute, Université catholique de Louvain, Brussels, Belgium
4   Laboratoire de Biologie Moléculaire Eucaryote, Centre de Biologie Intégrative (CBI), CNRS, UPS, Université de Toulouse, Toulouse, France
5   EA 7327, Université Paris Descartes, Sorbonne Paris-Cité, Paris, France
6   Laboratoire de Microbiologie clinique & Unité d'Immunologie, Hématologie et Rhumatologie Pédiatriques, AP-HP, Hôpital Necker, Enfants Malades, Paris, France
7   Pediatric Hematology, Faculty of Medicine, Baskent University, Ankara, Turkey
8   Pediatric Infectious Diseases, Department of Pediatric Infectious Diseases, Pamukkale University Medical Faculty, Denizli, Turkey
9   INSERM, UMR 1163, Genomics platform, Imagine Institute, Paris Descartes–Sorbonne Paris Cité University, Paris, France
10  Genomic Core Facility, Imagine Institute-Structure Fédérative de Recherche Necker, INSERM U1163, Paris, France
11  Centre d'Ingénierie Génétique Murine, Institut Pasteur, Paris, France
12  Muséum National d'Histoire Naturelle, UMR CNRS 7590, Institut de Minéralogie, de Physique des Matériaux et de Cosmochimie, IMPMC, Sorbonne Université, Paris, France
    *Corresponding author. Tel: +32-(0)2-7647574; E-mail: anabelle.decottignies@uclouvain.be
    **Corresponding author. Tel: +33-14-2754292; E-mail: patrick.revy@inserm.fr
    †These authors contributed equally to this work as senior authors

(Savage, 2014; Glousker *et al*, 2015). HH syndrome and Revesz syndrome are rare disorders that represent the most severe clinical variants of DC (Alter *et al*, 2012; Glousker *et al*, 2015). HH syndrome is characterized by early-onset bone marrow failure, intrauterine growth retardation, microcephaly and/or cerebellar hypoplasia, and other developmental defects (Glousker *et al*, 2015). Most HH patients die in their first decade because of severe infections as a consequence of profound immunodeficiency. To date, six genetic causes of HH have been reported, including germline mutations in the genes coding for the telomerase factors dyskerin (DKC1) and hTERT, for the shelterin components TIN2 and TPP1, and for the DNA helicase RTEL1 (Glousker *et al*, 2015). Recently, biallelic mutations in the PARN (poly(A)-specific ribonuclease)-encoding gene were reported by four independent laboratories in a total of nine patients as the sixth identified molecular cause of HH (Dhanraj *et al*, 2015; Moon *et al*, 2015; Tummala *et al*, 2015; Burris *et al*, 2016).

PARN is a ribonuclease from the DEDDh subfamily of nucleases that deadenylates poly(A) tails of RNA and therefore participates in the controlling of mRNA stability and gene expression (Balatsos *et al*, 2012). Moreover, two independent laboratories recently identified PARN as a factor involved in rRNA biogenesis in cultured human cell lines after PARN depletion by siRNA (Ishikawa *et al*, 2017; Montellese *et al*, 2017). Additionally, PARN deadenylates other non-coding RNAs including miRNAs, piRNAs, scaRNAs, snoRNAs, and the human telomerase RNA component h*TR/TERC* (Berndt *et al*, 2012; Katoh *et al*, 2015; Moon *et al*, 2015; Nguyen *et al*, 2015; Tseng *et al*, 2015; Zhang *et al*, 2015; Shukla *et al*, 2016, 2019; Ishikawa *et al*, 2017; Montellese *et al*, 2017). PARN depletion was initially reported to reduce h*TR* stability by impacting its 3′-end maturation regulation (Moon *et al*, 2015; Nguyen *et al*, 2015; Tseng *et al*, 2015). Further work demonstrated that PARN, by counteracting PAPD5-mediated oligoadenylation, prevents 3′-to-5′ degradation of h*TR* by the exosome (Shukla *et al*, 2016). Moreover, upon PARN depletion, the residual h*TR* was mislocalized into cytoplasmic foci. As exosome inactivation rescued h*TR* localization into Cajal bodies of PARN-depleted cells, it was suggested that PARN is not directly involved in h*TR* localization into Cajal bodies but that the mislocalization results from an increased instability of h*TR* RNA in these cells (Shukla *et al*, 2016).

Further linking *PARN* mutations to telomere defects was the interesting observation that the mRNA levels of *TRF1, DKC1,* and *RTEL1* were significantly down-regulated in blood cells from four PARN-deficient patients compared to controls (Tummala *et al*, 2015). In the same study, siRNA-mediated PARN depletion in human cell lines down-regulated the stability of *DKC1, RTEL1,* and *TRF1* mRNAs. However, the impact of PARN depletion on the steady-state levels of these mRNAs was not investigated (Tummala *et al*, 2015). As unexpected results were obtained upon PARN knock-down (KD) in mouse myoblasts, where a decrease in transcript abundance could be associated with an increased stability of the affected mRNA (Lee *et al*, 2012), the question remains open as to whether PARN defects down-regulate the expression of human telomere-related genes. In favor of this hypothesis, PARN KD in mouse myoblasts was associated with a reduced abundance of *Terf1, Terf2,* and *Rtel1* gene transcripts that was, however, not associated with a decrease in their respective mRNA half-life (Lee *et al*, 2012). Intriguingly, in the same study, authors detected an up-regulation of *Terc* RNA upon PARN KD (Lee *et al*, 2012), suggesting some differences in PARN targets between human and mouse.

Another important target of human PARN is *p53* mRNA. In human cancer cells, PARN KD was associated with the stabilization of *p53* mRNA (Devany *et al*, 2013). This, again, was not observed in PARN-depleted mouse myoblasts (Lee *et al*, 2012). Recently, Shukla *et al* (2019) reported that the up-regulation of p53 protein levels resulted from the down-regulation of some specific *p53* mRNA-binding miRNAs upon PARN depletion in human cancer cells. Increased p53 levels could participate in the premature aging phenotype of PARN-deficient cells either directly or indirectly, through an impact on the expression of telomere-related genes. Indeed, mice expressing the p53$^{\Delta31}$ hyperactive form of p53 were found to be affected in their telomere metabolism through the down-regulation of *Terf1, Tinf2, Dkc1,* and *Rtel1* gene expression (Simeonova *et al*, 2013), raising the interesting hypothesis that the down-regulation of telomere-related genes in human *PARN* mutant cells may result from p53 up-regulation (Mason & Bessler, 2015). This hypothesis has however not been tested so far.

We here identified two unrelated HH individuals carrying novel biallelic *PARN* mutations. By using *PARN*-mutated cells from patients as well as a *PARN* knock-out human cell line generated by CRISPR/Cas9 and carrying an inducible complementing *PARN* allele, we examined the functional consequences of PARN deficiency on telomere length and stability, expression of telomere-related genes, and rRNA processing. We also evaluated the requirement for p53 in the deregulation of telomere-related gene expression in cells lacking PARN. Furthermore, a *Parn* KO mouse model generated by CRISPR/Cas9 technology indicated that Parn is an essential factor in mice.

## Results

### Clinical features of two unrelated individuals

Individual 1 (P1) was born to a consanguineous family. She had an older sister who died from unknown cause at 2 years of age (Fig 1A). P1 was admitted to hospital at the age of 9 with pancytopenia, cerebellar ataxic gait, microcephaly, cerebellar hypoplasia, coarse hair, and dystrophic nails (Table 1 and Appendix Fig S1). Circulating B and NK lymphocytes were virtually absent, and bone marrow aspirate revealed hypocellular sample with very few hematopoietic stem cells and dysplastic megakaryocytes (Table 2). At 11 years old, P1 had no history of severe infection and did not require any blood or platelet transfusions.

Individual 2 (P2) was born to an unrelated non-consanguineous family (Fig 1A). She suffered from intrauterine growth retardation (IUGR) and, at birth, presented with hypotrophy (height and weight below −2 standard deviation (SD)) and major microcephaly (below −6 SD) (Table 1). Cerebral MRI, performed because of cerebellar ataxia and developmental delay, revealed cerebellar vermian and hemispheric atrophy and atrophy of the pons. Since the first months of life, she presented with feeding troubles (without other gastrointestinal symptoms or oral leukoplakia), and recurrent mild upper and lower respiratory tract infections. Thrombopenia was diagnosed at 6 months old, with bone marrow biopsy showing severe hypoplasia and dysplasia of the megakaryocytic lineage. At the age of

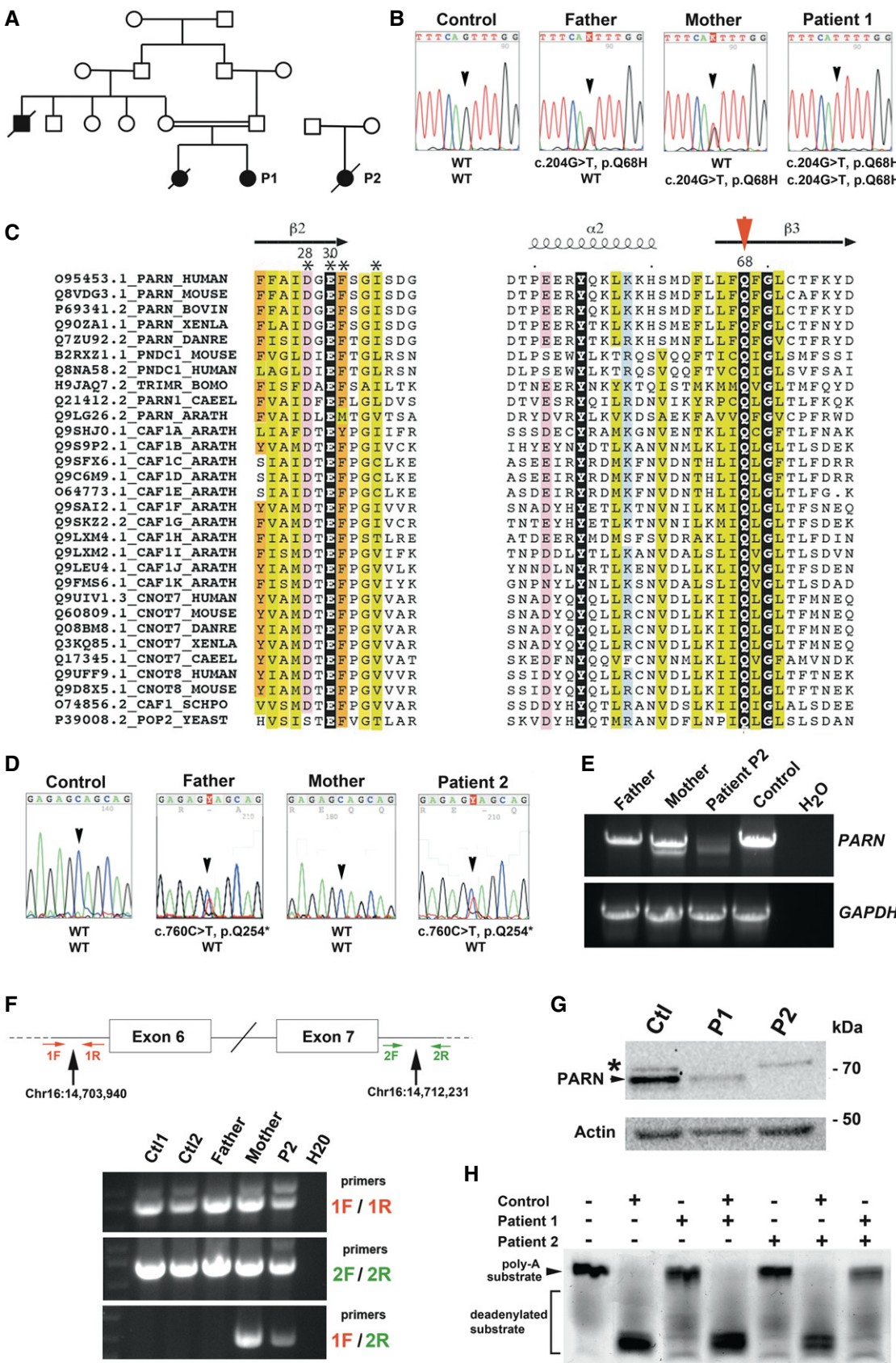

**Figure 1.**

**Figure 1. Identification of *PARN* mutations in two unrelated HH patients.**

A  Pedigree of Patients 1 and 2 and their families.
B  Direct sequencing of *PARN* in a control, P1, and her parents.
C  Alignment of the two first conserved blocks of sequences from the DEDDh subfamily of nucleases (UniProt accession numbers). Conserved residues are colored, and the conserved glutamine is highlighted with a red arrowhead. Secondary structures are reported on top, as observed in the experimental 3D structure of human PARN (pdb 2A1R) (Wu *et al*, 2005).
D  Direct sequencing of *PARN* in a control, P2, and her parents.
E  Aberrant splicing products detected in P2 and her mother but not in P2′s father nor in a healthy control. *GAPDH* RT–PCR was performed as control.
F  Detection of the exon 6/7 deletion by PCR with specific primers (1F/2R).
G  PARN detection in cell lysates from P1 and P2 and a healthy control. Actin was used as loading control.
H  *In vitro* deadenylation activity using protein extracts from control and P1 and P2 SV40T-transformed fibroblasts. Mixing protein extracts from control together with P1 or P2 eliminated the deadenylation defect, whereas pairwise mixing of P1 and P2 failed to complement the deadenylation defect.

**Table 1. Clinical features of patients.**

|  | Patient 1 | Patient 2 |
|---|---|---|
| Sex | Female | Female |
| Consanguinity | Yes | No |
| **Developmental features** | | |
| IGR | NA | Yes |
| Prematurity | NA | No (37WG) |
| Hypotrophy | | Yes (< −2 SD) |
| Dysmorphy | Yes (coarse hair, dystrophic nails) | Yes (short neck, protruding ears, widely spaced and inverted nipples) |
| Aplastic anemia | Yes | Yes, progressive |
| Immunodeficiency | Yes | Yes |
| **Neurological features** | | |
| Microcephaly | Yes | Yes (< −6 SD) |
| Cerebellar atrophy | Yes | Yes |
| Other | No | Atrophy of the pons |
| Gastrointestinal features | No | Feeding troubles since the neonatal period |
| Skin features | No | No |
| Outcome | Alive at 11 years of age without severe infection | Death at 3 years of age after severe infection |

NA, not available.

10 months, she showed moderate lymphopenia, predominantly on naïve CD4 T cells and B cells (Table 2). Serum immunoglobulin levels were normal at the age of 6 and 10 months. P2 rapidly developed progressive pancytopenia, requiring blood and platelet transfusions since the age of 22 months. She died from severe bacterial infection at 3 years old.

Overall, P1 and P2 exhibited clinical features akin to Høyeraal–Hreidarsson syndrome (Glousker *et al*, 2015).

### Identification of *PARN* mutations in individuals P1 and P2

In P1, the combined analyses of whole genome homozygosity mapping (WGHM) and whole exome sequencing (WES) focused on homozygous genetic variants absent from dbSNPs, EVS, 1,000 genome and from our in-house databases (8,319 individuals) and located in chromosomal regions co-segregating with the disease identified homozygous mutations in the Titin-encoding gene (NM_001267550.1: c.9413C>A; Chr2(GRCh37):g.179632544G>T; p.Ala3138Glu) and in the PARN-encoding gene (NM_002582.2:c.204G>T; Chr16(GRCh37): g.14721167C>A; p.Gln68His). We considered *PARN* gene as the strongest candidate because biallelic *PARN* mutations had been recently reported in HH (Dhanraj *et al*, 2015; Moon *et al*, 2015; Tummala *et al*, 2015; Burris *et al*, 2016). Sanger sequencing confirmed the homozygous c.204G>T *PARN* mutation in P1, with both parents being heterozygous for the mutation (Fig 1B). This variant was not listed in gnomAD database (http://gnomad.broadinstitute.org/), further supporting its deleterious effect. Interestingly, even though overall sequence identities are generally low, p.Gln68 (Q68) is highly conserved in the DEDDh subfamily of nucleases (Fig 1C) and p.Gln68His mutation is predicted to disturb catalytic activity (Appendix Fig S2).

In P2, WES analysis revealed a heterozygous *PARN* mutation leading to a premature stop codon (NM_002582.2:c.760C>T, Chr16 (GRCh37):g.14698026G>A, p.Gln254*). This variant, absent from gnomAD database, was inherited from her father (Fig 1D). No *PARN* mutation was detected on the second allele, but we detected several distinct RT–PCR products for *PARN* in both P2 and her mother, along with a reduced abundance of P2′s RT–PCR products (Fig 1E). The heterogeneity in RT–PCR products was due to complex splice aberrations including the skipping of one or more exons (Appendix Fig S3A). A capture library was generated by using a BAC covering the whole *PARN* gene sequence, followed by high-throughput sequencing. Coverage analysis revealed a twofold reduction in the abundance of the genomic region spanning 8,293 base pairs (Chr16:14,703,940–14,712,231) comprising the *PARN* exons 6 and 7 in the mother and the patient's cells (Appendix Fig S3B). The heterozygous deletion was then confirmed by PCR (Fig 1F) and Sanger sequencing (Appendix Fig S3C).

At the protein level, PARN was undetectable from lysates of P2′s SV40T-transformed fibroblasts and was also strongly reduced in cell extracts from P1 (Fig 1G). As expected, PARN down-regulation was associated with a strong impairment of *in vitro* deadenylation activity on a polyadenylated RNA substrate (Fig 1H). Additionally, mixing P1 and P2 cell extracts did not rescue the deadenylation activity (Fig 1H), further supporting the notion that the molecular defect was identical in P1 and P2.

Intriguingly, individual P1, although presenting with severe clinical features, did not exhibit severe infection, profound anemia, or thrombocytopenia requiring blood transfusion, as observed in most HH cases. Sequencing analysis in a recent P1's blood sample did not

**Table 2. Immunological features of patients.**

| | Patient 1 | | Patient 2 | | |
|---|---|---|---|---|---|
| Age | 9 years | 11 years | 6 months | 10 months | 23 months |
| White cell count—×10$^9$/l | 2.7 (5.5–15.5) | 3.3 (5.5–15.5) | 8.8 (6–17.5) | 6.7 (6–17.5) | 3.1 (6–17.5) |
| Polymorphonuclear neutrophils—×10$^9$/l | 1.1 (1.8–8.0) | 1.4 (1.8–8.0) | 3.0 (1.5–8.5) | 3.2 (1.5–8.5) | 0.8 (1.5–8.5) |
| Lymphocytes—×10$^9$/l | NA | NA | 4.4 (3.0–9.5) | 2.3 (3.0–9.5) | 2.2 (3.0–9.5) |
| Hemoglobin—g/dl | 10 (11.5–13.5) | 7.1 (11.5–13.5) | 11.9 (10.5–12.0) | 12.1 (10.5–12.0) | 7.3 (10.5–12.0) |
| Platelets—×10$^9$/l | 29 (175–420) | 37 (175–420) | 67 (175–500) | 38 (175–500) | 5 (175–500) |
| Reticular platelets—% | NA | NA | NA | 6.5 (1.0–7.0) | NA |
| T cells—% (normal range) | | | | | |
| CD3$^+$ | 88 (60–76) | 92 (60–76) | 73 (49–76) | 84 (49–76) | NA |
| CD4$^+$ | 47 (31–47) | 31 (31–47) | 40 (31–56) | 41 (31–56) | NA |
| CD31$^+$CD45RA/CD4$^+$ | 27 (43–55) | NA | NA | 37 (60–72) | NA |
| CD8$^+$ | 41 (18–35) | 59 (18–35) | 15 (12–24) | 33 (12–24) | NA |
| B CD19$^+$ cells – % | 0.1 (13–27) | 2 (13–27) | 19 (14–37) | 12 (14–37) | NA |
| Natural killer CD16$^+$CD56$^+$ cells — % | 0.1 (4–17) | NA | 7 (3–15) | 4 (3–15) | NA |
| Serum immunoglobulins level: g/l | | | | | |
| IgG | NA | 9.76 (6.55–12.29) | 3.35 (3.35–6.23) | 5.30 (3.35–6.23) | NA |
| IgA | NA | 1.74 (0.5–2.03) | 0.41 (0.27–0.86) | 0.43 (0.27–0.86) | NA |
| IgM | NA | 1.21 (0.53–1.62) | 1.27 (0.48–1.36) | 0.52 (0.48–1.36) | NA |

For all values, normal range or normal thresholds are indicated in brackets. Abnormal low values are highlighted in bold.
NA, not available.

reveal any somatic genetic modification in *PARN* (data not shown), ruling out the possibility of a spontaneous genetic reversion or correction as reported in other hematologic genetic disease (Le Guen *et al*, 2015; Tesi *et al*, 2017). Somatic *TERT* promoter-activating mutations in blood cells leading to increased telomerase expression can also counteract the deleterious effect of inherited heterozygous loss-of-function mutations in h*TERT,* h*TERC,* or *PARN* (Maryoung *et al*, 2017; Gutierrez-Rodrigues *et al*, 2018). However, sequencing analysis of *TERT* promoter in P1's blood cells did not either reveal any variant that could explain the relative mild hematologic phenotype (data not shown).

## Telomere length defect and telomere instability in PARN-mutated cells from P1 and P2

We first assessed whether PARN deficiency in patients was accompanied by a telomere length defect as previously reported (Dhanraj *et al*, 2015; Moon *et al*, 2015; Tummala *et al*, 2015; Burris *et al*, 2016). Telomere restriction fragment (TRF) measurement revealed abnormally short telomeres in peripheral blood mononuclear cells from P1 and P2 as compared to their parents (Fig 2A). Quantitative telomeric FISH (qTelo-FISH) on patients' SV40T fibroblasts, using Muntjac cells to normalize the fluorescence intensity of telomeric FISH signal (Zou *et al*, 2002), confirmed the short telomere phenotype in P1 and P2 (Fig 2B–D). The short telomere phenotype observed in patients' cells was accompanied by telomere dysfunction, detected by the presence of DNA repair factor 53BP1 at telomeres, known as TIF (telomere dysfunction-induced foci) (d'Adda di Fagagna *et al*, 2003; Takai *et al*, 2003; Fig 2E and F). Accordingly,

primary fibroblasts from P1 showed a significant increase in cellular senescence, as assessed by the senescence-associated (SA)–β-galactosidase activity assay, when compared to two healthy controls at similar passage (Fig 2G). Due to a severe growth defect of P2's primary fibroblasts, TIF and senescence analyses could not be performed in these cells.

Next, we assessed whether the telomere length defect of P1 and P2 was associated with any telomere instability. Telomeric FISH performed on metaphase spreads of SV40T-transformed fibroblasts at early passages revealed a statistically significant increase in terminal deletions (Fig 2H and I) and telomere sister loss (Fig 2H–J) in both patients' cells, while multiple telomeric signals and telomere fusions were not overrepresented (not shown). These results suggested that, in addition to accelerated telomere shortening, PARN-mutated cells from patients exhibited increased telomere instability.

## PARN regulates the expression levels of telomere-related gene transcripts, but not TERRA

In light of the possible impact of PARN on mRNA levels of shelterin genes (Tummala *et al*, 2015), the increased telomere instability observed in P1 and P2's cells may be related to a down-regulation of shelterin gene expression. Previous work from Tummala *et al* (2015) reported that transient depletion of human *PARN* was associated with a decreased stability of *DKC1, RTEL1,* and *TRF1* transcripts, but a comparison of the steady-state mRNA levels was not provided. To assess the impact of stable PARN depletion on the expression of telomere-related genes and to perform functional

studies on telomeres from these cells, we generated PARN knockout (KO) mutants in the telomerase- and p53-positive human cell line HT1080 through CRISPR/Cas9 mutagenesis. We obtained a PARN KO clone (HT1080$^{PARN\ KO}$) carrying *PARN* frameshift

mutations on both alleles (1 bp deletion on one allele and 1bp insertion on the other, Appendix Fig S4), leading to a complete loss of PARN protein (Fig 3A). Accordingly, HT1080$^{PARN\ KO}$ cells exhibited a defective *in vitro* deadenylation activity on a poly-(A) RNA

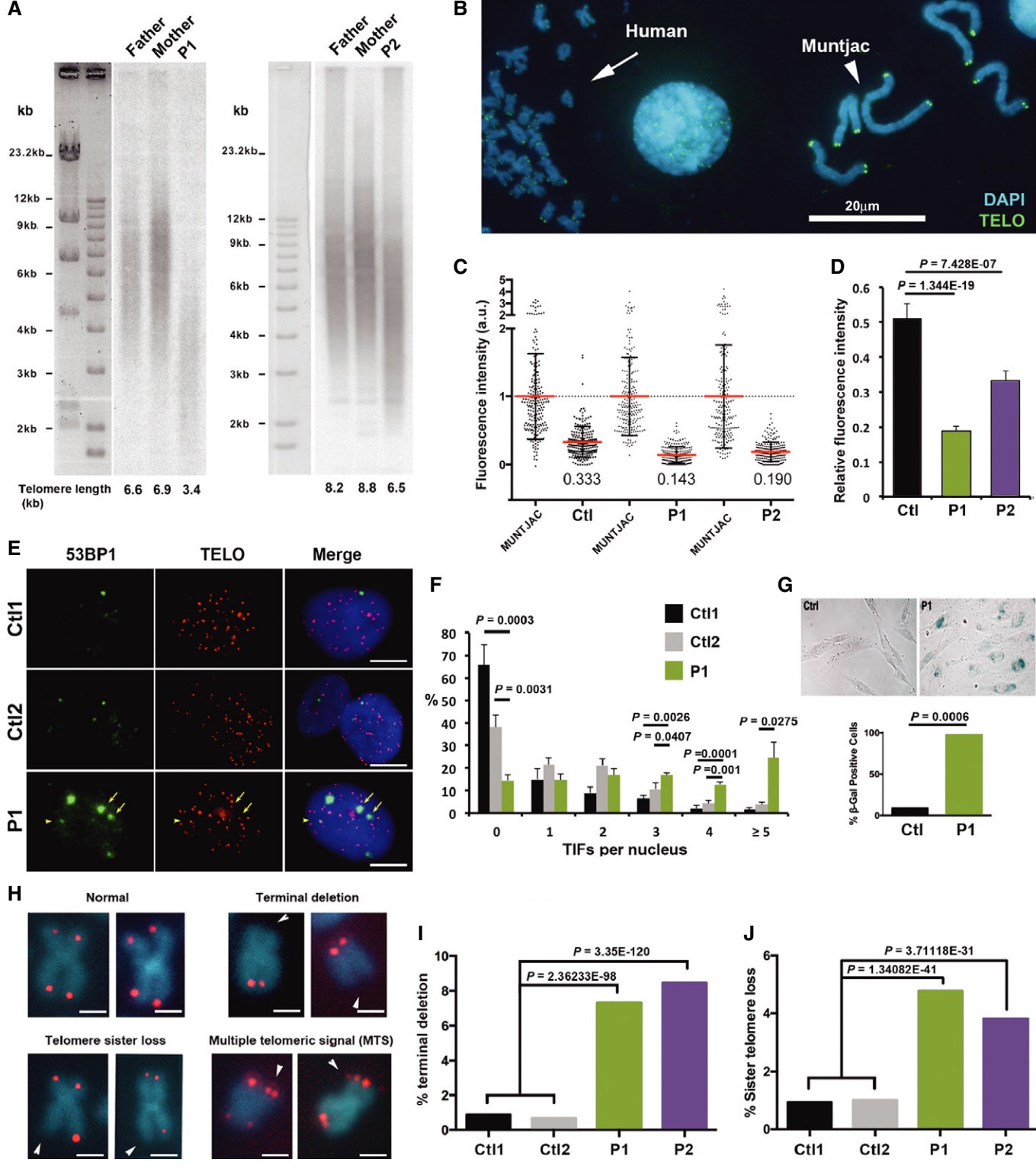

Figure 2.

**Figure 2.  Short and dysfunctional telomeres in cells from HH patients.**

A  Mean telomere length (kb) of whole blood cells from patients and their parents estimated with the TRF method.

B  Representative picture of telomeric signals used for Q-FISH analysis on metaphase spreads from SV40T-transformed fibroblasts and Muntjac cells used to normalize the signals.

C  Individual and mean values of Q-FISH analyses for control and patients. Mean fluorescence intensity of Muntjac cells was set to 1. Error bars indicate s.d.

D  Graphical representation of (C) showing the average of fluorescence ratios (relative to Muntjac). Error bars indicate s.e.m. The non-parametric Kruskal–Wallis test was applied to compare relative fluorescence values between Ctl and P1, and Ctl and P2 3 independent experiments.

E  Representative pictures of nuclei from two controls and patient's primary fibroblasts showing 53BP1 foci (green) and telomeres (red, detected by Telo-FISH). Yellow arrows indicate telomere dysfunction-induced foci (TIF). The scale bar corresponds to 5 μm.

F  Quantification of (E) in primary fibroblasts from two healthy controls (passages > 4) and from Patient 1 (passages < 4). Control 1: $n$ = 224; Control 2: $n$ = 263; Patient 1: $n$ = 231. Averages are shown, and error bars indicate s.e.m.. Unpaired Student's $t$-tests were applied when indicated.

G  Representative pictures of SA–β-galactosidase staining in control (passage 15) and P1's primary fibroblasts (passage 6). Results are expressed as the percentage of SA–β-galactosidase-positive cells (averages, lower panel). Control: $n$ = 436; P1: $n$ = 436. A test to compare two population proportions was applied.

H  Representative pictures of chromosomes with normal or aberrant telomeres detected by Telo-FISH. The scale bar corresponds to 1 μm. Arrowheads indicate chromatid ends lacking telomeric signal.

I, J  Quantification of terminal deletions (I) and sister telomere losses (J) from three independent experiments (counted chromatids: Ctl1: $n$ = 2,224; Ctl2: $n$ = 5,696; P1: $n$ = 4,764; P2: $n$ = 7,908). Averages are shown, and chi-square tests were applied to compare Ctl1/2 with either P1 or P2.

substrate (Fig 3B) and reduced telomerase activity (Fig 3C) associated with a down-regulation of h*TR* transcripts (Fig 3D). Additionally, as recently reported in PARN knock-down cells (Shukla *et al*, 2016), RNA-FISH revealed a mislocalization of h*TR* to the cytoplasm of HT1080[PARN KO] cells (Fig 3E). Next, HT1080[PARN KO] cells were transduced with either control pCW57 plasmid encoding GFP ("empty") or pCW57-PARN plasmid carrying an inducible WT PARN coding sequence ("indPARN"). Doxycycline-induced expression of PARN (Fig 3F) was found to complement the defective deadenylation activity of HT1080[PARN KO] cells (Fig 3G). For sake of simplicity, cell lines will be denominated as follows: WT/empty, WT/PARN, KO/empty, and KO/PARN.

qRT–PCR experiments confirmed the involvement of PARN in h*TR* RNA transcript abundance ($P$=0.001) and supported the PARN-dependent impact on telomere-related gene transcripts since the complementation of HT1080[PARN KO] cells by WT PARN rescued the transcript levels of *TRF1* ($P$ = 0.003), *TRF2* ($P$ = 0.0002), *POT1* ($P$ = 0.001), *DKC1* ($P$ = 0.05), and *TPP1* ($P$ = 0.0004) found to be down-regulated in KO/empty cells (Fig 3H). In our system, the impact of PARN KO was the strongest for TRF2, with a more than twofold reduction in transcript abundance associated with a down-regulation at the protein level (Fig 3H and I). Although differences did not reach statistical significance, *RAP1* and *RTEL1* transcripts were also down-regulated in KO/empty cells (Fig 3H).

In agreement with previous reports showing that PARN depletion increases *p53* mRNA levels by ~1.7–1.9-fold (Zhang & Yan, 2015; Shukla *et al*, 2019) *p53* transcript levels were increased by about twofold in KO/empty cells ($P$ = 0.001) (Fig 3J). Up-regulation at the protein level was of about twofold as well (Fig 3I). Surprisingly however, p53 levels (mRNA or protein) were not reduced back in PARN-complemented cells (Fig 3I and J). Similar results were obtained for *p21* mRNA levels that displayed a more than twofold increase in KO/empty cells ($P$ = 0.0009) but failed to return to WT levels after PARN complementation (Fig 3J). These observations were reminiscent to the previously proposed adaptation of PARN-depleted cells through the up-regulation of other cellular deadenylases (Zhang & Yan, 2015).

TElomeric Repeat-containing RNAs (TERRA), the telomeric non-coding RNAs transcribed from subtelomeric promoters, also participate in telomerase regulation and telomere protection (Azzalin *et al*, 2007; Chu *et al*, 2017). Because PARN deadenylase substrates include other non-coding RNAs like miRNAs, piRNAs, scaRNAs, snoRNAs, we next examined whether PARN depletion may affect TERRA levels. To avoid any bias due to distinct telomere lengths, previously found to affect TERRA abundance (Arnoult *et al*, 2012), we quantified TERRA molecules produced from various chromosome ends in HT1080[PARN KO] cells complemented or not with PARN. Our qRT–PCR analyses did not however reveal any impact of PARN on TERRA levels (Fig 3K).

**Figure 3.  PARN KO cells display reduced telomerase activity and down-regulation of shelterin and telomere-related gene expression.**

A  Validation of PARN KO in HT1080 cells by Western blot. GAPDH is used as loading control.

B  *In vitro* deadenylation activity assay using protein extracts from HT1080 (WT) and HT1080[PARN KO] (PARN KO) cells.

C  TRAP assay using successive dilutions (500, 250, 125, and 62.5 ng) of cell extracts from WT or PARN KO cells. An internal control (IC) for PCR was used.

D  qRT–PCR analysis of h*TR* expression in WT and PARN KO cells. Expression levels were normalized first to *ACTB* and then to WT. Three independent RNA extractions were performed for each cell line. Error bars indicate s.e.m.

E  Representative pictures of FISH against h*TR* in WT and PARN KO cells. Arrows indicate h*TR* foci. Scale bar 5 μm.

F  PARN detection by Western blot in the indicated conditions. Expression was induced by incubating cells with 10 ng/ml doxycycline for 72 h. Actin is used as loading control. PARN/actin ratios, normalized to doxycycline-treated WT/PARN cells, are shown below.

G  *In vitro* deadenylation activity assay using protein extracts from doxycycline-treated WT/empty, WT/PARN, KO/empty, and KO/PARN cells.

H  qRT–PCR analysis of the indicated gene transcripts in doxycycline-treated WT/empty, WT/PARN, KO/empty, and KO/PARN cells. Expression levels were normalized first to *ACTB* and then to WT/empty. Three independent doxycycline inductions were performed for each cell line. Averages are shown, and error bars indicate s.e.m. Unpaired Student's $t$-tests were applied.

I  Representative Western blot analysis of PARN, TRF2, and p53. Actin was used as loading control. Mean TRF2/actin levels from two independent experiments are indicated below.

J  Same as (H) for *p53* and *p21* transcripts. Averages are shown, and error bars indicate s.e.m. Unpaired Student's $t$-tests were applied. Three independent doxycycline inductions were performed for each cell line.

K  qRT–PCR analysis of TERRA from the indicated chromosome ends in doxycycline-treated KO/empty and KO/PARN cells. Expression levels were normalized first to *ACTB* and then to KO/empty. Three independent doxycycline inductions were performed for each cell line. Error bars indicate s.e.m.

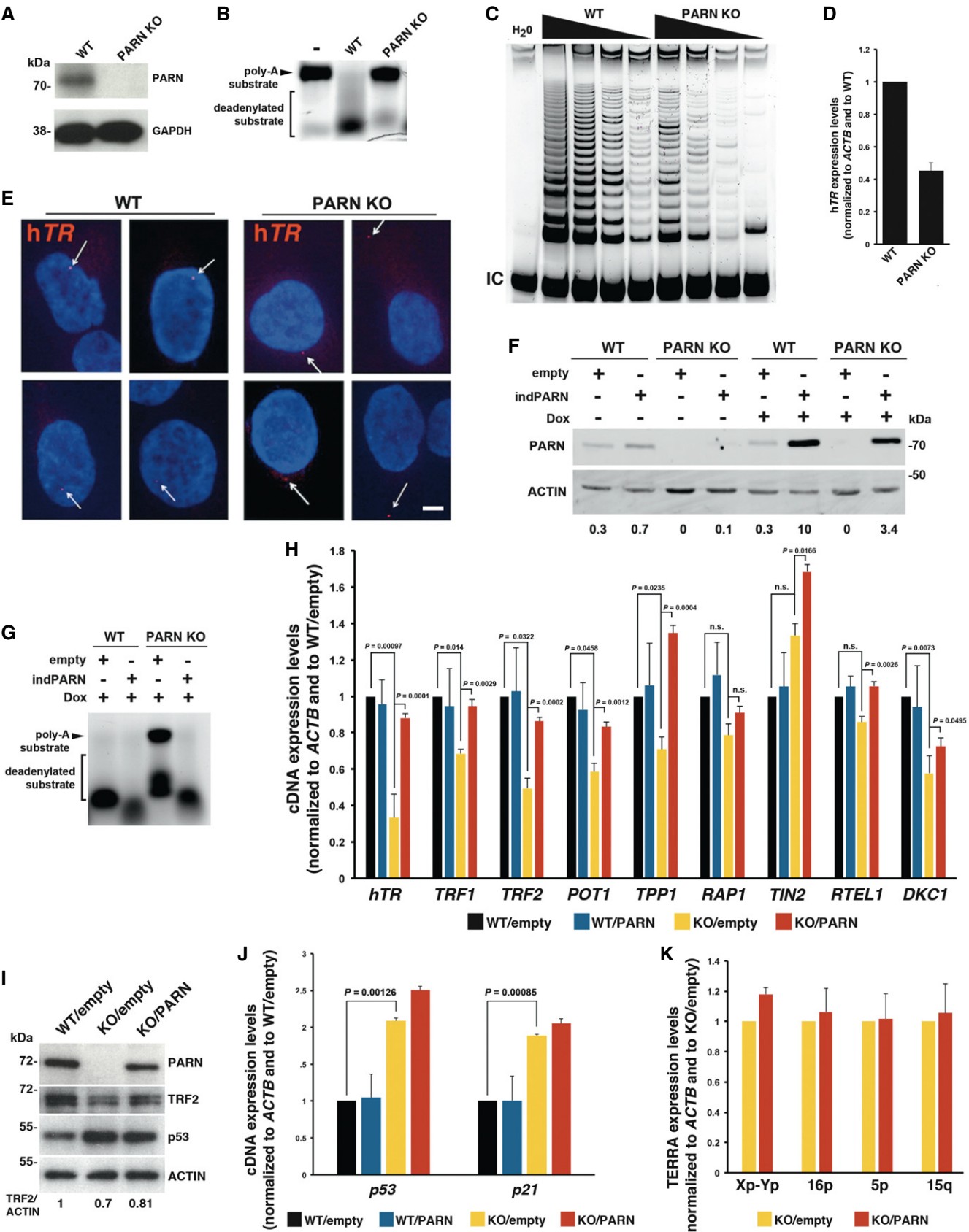

**Figure 3.**

## Telomere instability in HT1080<sup>PARN KO</sup> cells is independent of telomere length

As the defective expression of telomere-related genes may induce telomere instability, we next tested whether the telomere instability observed in PARN-deficient cells from patients (Fig 2I and J) was also detected in HT1080<sup>PARN KO</sup> cells. Accordingly, telomeric FISH revealed a significant increase in terminal deletions, telomere sister losses, telomere–telomere fusions, and multiple telomeric signals in KO/empty compared to WT/empty cells (Fig 4A). Strikingly, the telomeric aberrations were abolished after PARN induction via 72h of Dox treatment in HT1080<sup>PARN KO</sup> cells (Fig 4A). Since the induction of WT PARN during 72h was not sufficient to globally modify telomere length (Fig 4B), we concluded that the telomere instability of HT1080<sup>PARN KO</sup> cells was independent of telomere length. Therefore, this result suggests that PARN participates in telomere stability, possibly through the regulation of telomere-related gene expression.

## Down-regulation of *DKC1* transcripts in HT1080<sup>PARN KO</sup> cells is dependent on p53

Our observations that, despite the lack of *p53* mRNA rescue in PARN-complemented HT1080<sup>PARN KO</sup> cells, the expression levels of h*TR*, *TRF1*, *TRF2*, *POT1*, *TPP1*, *RAP1*, and *RTEL1* were restored

(Fig 3H) suggested that p53 was not involved in the down-regulation of these gene transcripts. On the contrary, *DKC1* transcript levels were not restored upon PARN induction in complemented cells (Fig 3H). To further investigate the involvement of p53 in *DKC1* down-regulation upon PARN depletion, we transiently knocked down p53 using siRNAs in WT/empty and KO/empty cells (Fig 5A). Although p53 knock-down did not suppress the down-regulation of h*TR*, *TRF1*, *TRF2*, *POT1*, *TPP1*, or *RTEL1* transcripts levels, *DKC1* mRNA levels were similar in WT/empty and KO/empty cells treated with sip53, suggesting that *DKC1* down-regulation is a consequence of p53 induction (Fig 5B). To verify whether this p53-dependent impact on *DKC1* transcripts may also apply to cells from HH patients, we compared *DKC1* levels in primary and SV40T-transformed fibroblasts from P1 (SV40T is known to deregulate p53 activity; Ahuja *et al*, 2005). Accordingly, we detected a strong up-regulation of *DKC1* transcript levels in SV40T fibroblasts of P1 patient (Fig 5C). Conversely, and further supporting their independence from p53, other transcripts, including h*TR* or *TRF1*, were not up-regulated in SV40T-transformed fibroblasts from P1 (Fig 5C).

## Defective ribosomal RNA biogenesis in PARN-deficient cells

Recently, PARN was identified as the exonuclease that trims the 3′ end of 18S-E pre-ribosomal RNA (pre-rRNA; Ishikawa *et al*, 2017;

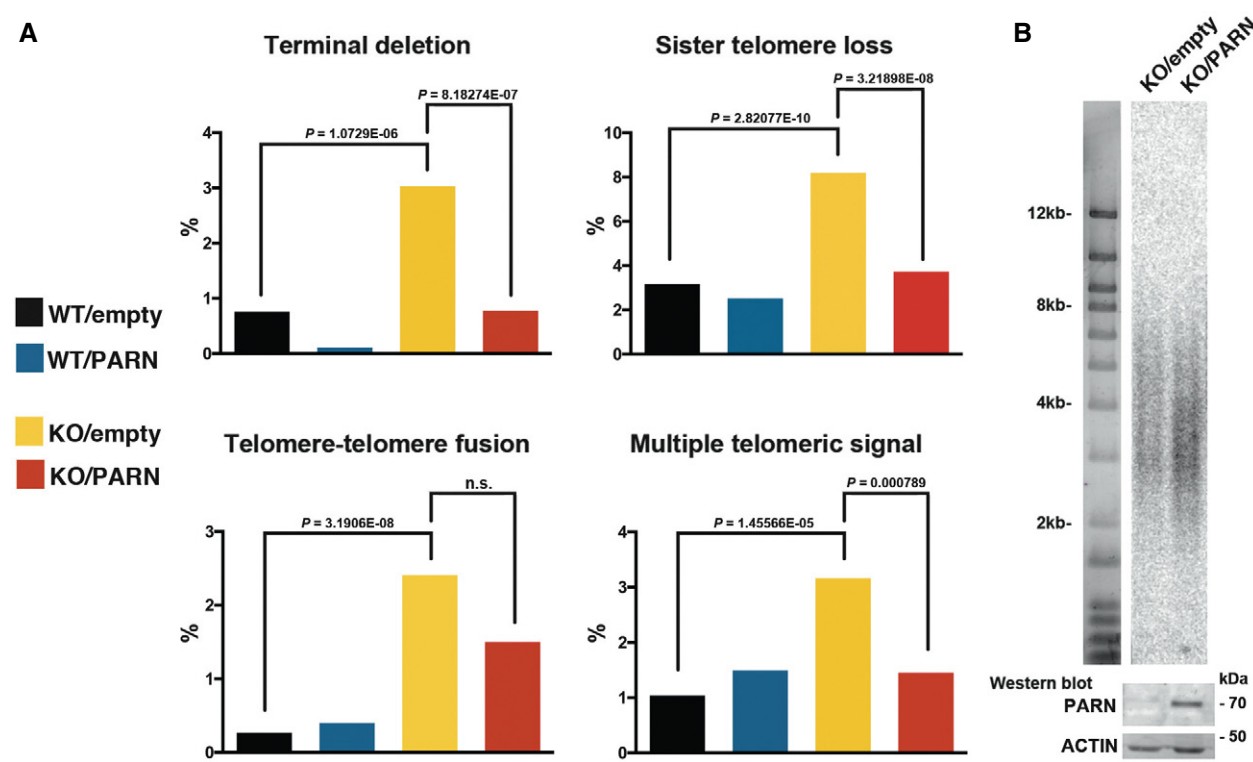

**Figure 4. PARN KO cells display telomere instability independently of telomere length.**

A   Analysis of telomeric aberrations by Telo-FISH in doxycycline-treated WT/empty, WT/PARN, KO/empty, and KO/PARN cells. Two independent experiments were performed. Counted chromatids: WT/empty: *n* = 1,824; WT/PARN: *n* = 1,744; KO/empty: *n* = 1,452; KO/PARN: *n* = 1,920. Averages are shown, and chi-square tests were applied to compare KO/empty with either WT/empty or KO/PARN.

B   TRF analysis of telomere length (kb) in KO/empty and KO/PARN cells treated with 10 ng/ml Dox for 72 h. A control Western blot with PARN and actin is shown below for the corresponding samples.

Montellese *et al*, 2017), the last precursor to the 18S rRNA. The full-length 18S-E pre-rRNA (18S-E$_{FL}$) generated by endonucleolytic cleavage by hUTP24 at site E is then submitted to exonucleolytic trimming by PARN before export to the cytoplasm (Appendix Fig S5; Wells *et al*, 2016). Interestingly, we found that rRNA biogenesis is impaired in patients with a marked accumulation of 18S-E pre-rRNAs in P1 and P2's cells relative to control cells (Fig 6A–D, 5′ITS1 probe, and Appendix Fig S5) Moreover, as expected from PARN loss of function, untrimmed 18S-E$_{FL}$ precursors were detected in P1 and, to a lesser extent, in P2's cells, while they were hardly detected in control cells (Fig 6C and D, ITS1-59 probe). The 18S/28S ratios were unchanged in patient cells despite this deficient processing, consistent with previous observation that the 3′ end of 18S rRNA precursors unprocessed by PARN can still be matured, albeit less efficiently, by endonuclease NOB1. As previously reported for PARN-depleted cell lines (Ishikawa *et al*, 2017; Montellese *et al*, 2017), the processing defect in P1 and P2's cells only impacted the 18S rRNA pathway, but did not affect 5.8S nor 28S precursors (ITS2 probes). A closer examination further revealed an accumulation of 30S$^{+1}$ pre-rRNAs in P1 (5′ETS probe), while these precursors were very sparse in control and P2 (Fig 6C). This defect in very early cleavage at site A′ was not observed upon the mere knock-out of PARN, and could correspond to a specific impact of the homozygous Gln68His PARN mutation, which remains to be investigated.

rRNA biogenesis was also affected in HT1080$^{PARN\ KO}$ relative to HT1080$^{PARN\ WT}$ cells as revealed by a sharp increase in 18S-E and 18S-E$_{FL}$ pre-rRNAs (Fig 6E and G). Importantly, induction of WT PARN expression complemented the defect observed in HT1080$^{PARN\ KO}$ cells, with a strong reduction of 18S-E and 18S-E$_{FL}$ precursors to the levels observed in HT1080$^{PARN\ WT}$ cells (Fig 6F and G). The rescue of defective rRNA biogenesis in PARN-complemented HT1080$^{PARN\ KO}$ cells that, as detailed earlier, keep high levels of p53 further suggests that the impact of PARN depletion on rRNA biogenesis occurs independently of p53.

## Early embryonic lethality of *Parn* KO mice

To investigate the consequences of PARN deficiency *in vivo*, we generated a *Parn* KO mouse model by using CRISPR/Cas9 technology. We selected two F0-derived F1 mice, Parn#25$^{+/-}$ and Parn#29$^{+/-}$, that carried, respectively, heterozygous 1 bp deletion and 2 bp deletion in the *Parn* coding sequence, both predicted to generate frameshift and premature stop codon (Appendix Fig S6). Parn #25$^{+/-}$ and Parn#29$^{+/-}$ were crossed onto C57BL/6 to segregate the CRISPR/Cas9-generated mutant alleles. SV40-transformed mouse embryonic fibroblasts (MEFs) obtained from Parn#25$^{+/-}$ confirmed the twofold reduction in Parn protein levels (Fig 7A). Interestingly, these MEFs also exhibited a strong accumulation of 18S-E pre-rRNAs (Fig 7B and C, 5′ITS1 probe). Unlike what was observed in human cells however, Parn haploinsufficiency also induced a moderate accumulation of 12S pre-rRNAs, along with slightly shorter precursors (ITS2 probes). This could be indicative of an involvement of mouse PARN in 12S pre-rRNA early processing or a quality control process, not found in human cells (Ishikawa *et al*, 2017; Montellese *et al*, 2017). Altogether, our data attest for a biological impact of Parn haploinsufficiency in MEFs and suggest a conserved function for mouse Parn in 18S rRNA biogenesis.

Parn#25$^{+/-}$ or Parn#29$^{+/-}$ mice were interbred to obtain *Parn*$^{-/-}$ animals. However, crosses yielded no homozygous null pup (Fig 7D; 96 animals screened; $P = 1.5E-08$, chi-square test) and we could not either detect any null embryo at E11.5 (Fig 7E; $n = 15$ screened; $P = 0.002535$, chi-square test), arguing for an early embryonic lethality that occurs too early in development for cell lines to be derived. PCR amplification and direct sequencing in *Parn*$^{+/-}$ heterozygous animals of the seven genomic loci predicted to be putatively off-targeted did not reveal any mutation, ruling out any off-target effect (Appendix Table S1). Since p53 deficiency was previously reported to rescue the deleterious effects of aberrant short telomeres (Chin *et al*, 1999), we next tested whether the

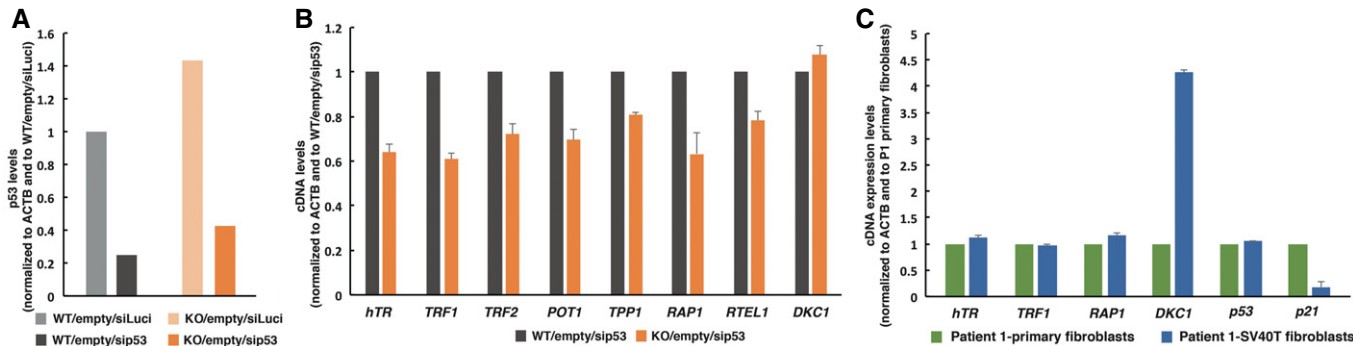

**Figure 5. Down-regulation of *DKC1* mRNA in PARN KO cells is dependent on p53.**

A   HT1080 (WT) and HT1080$^{PARN\ KO}$ (KO) cells stably transfected with GFP plasmid (empty) were transfected with siRNAs against *p53* for 72h before RNA extraction and qRT–PCR. siRNAs against luciferase (siLuci) were used as control. *p53* expression levels were normalized first to *ACTB* and then to WT/empty/siLuci cells. Experiment was performed in triplicate. Error bars indicate s.e.m.

B   Cells were treated as in (A), and transcript levels of the indicated genes were measured by qRT–PCR. cDNA expression levels were normalized first to *ACTB* and then to WT/empty/siLuci cells. Experiment was performed in triplicate. Error bars indicate s.e.m.

C   qRT–PCR analyses of the indicated genes were performed on RNA extracted from P1's primary or SV40T-transformed fibroblasts. cDNA expression levels were normalized to primary fibroblasts. Two independent RNA extractions were performed. Error bars indicate s.e.m.

lack of p53 could abolish the embryonic lethality of Parn-deficient embryos. However, the crossing of $Parn^{+/-}$ $p53^{+/-}$ animals with either $Parn^{+/-}$ $p53^{+/-}$ (Fig 7F) or $Parn^{+/-}$ $p53^{-/-}$ mice (Fig 7G) did not retrieve any mouse deficient for both p53 and Parn (a total of 105 animals were screened; $P = 0.000886$, chi-square test). Thus, these results indicated that Parn deficiency in mouse leads to embryonic lethality before stage E11.5 that is not overcome by the absence of p53.

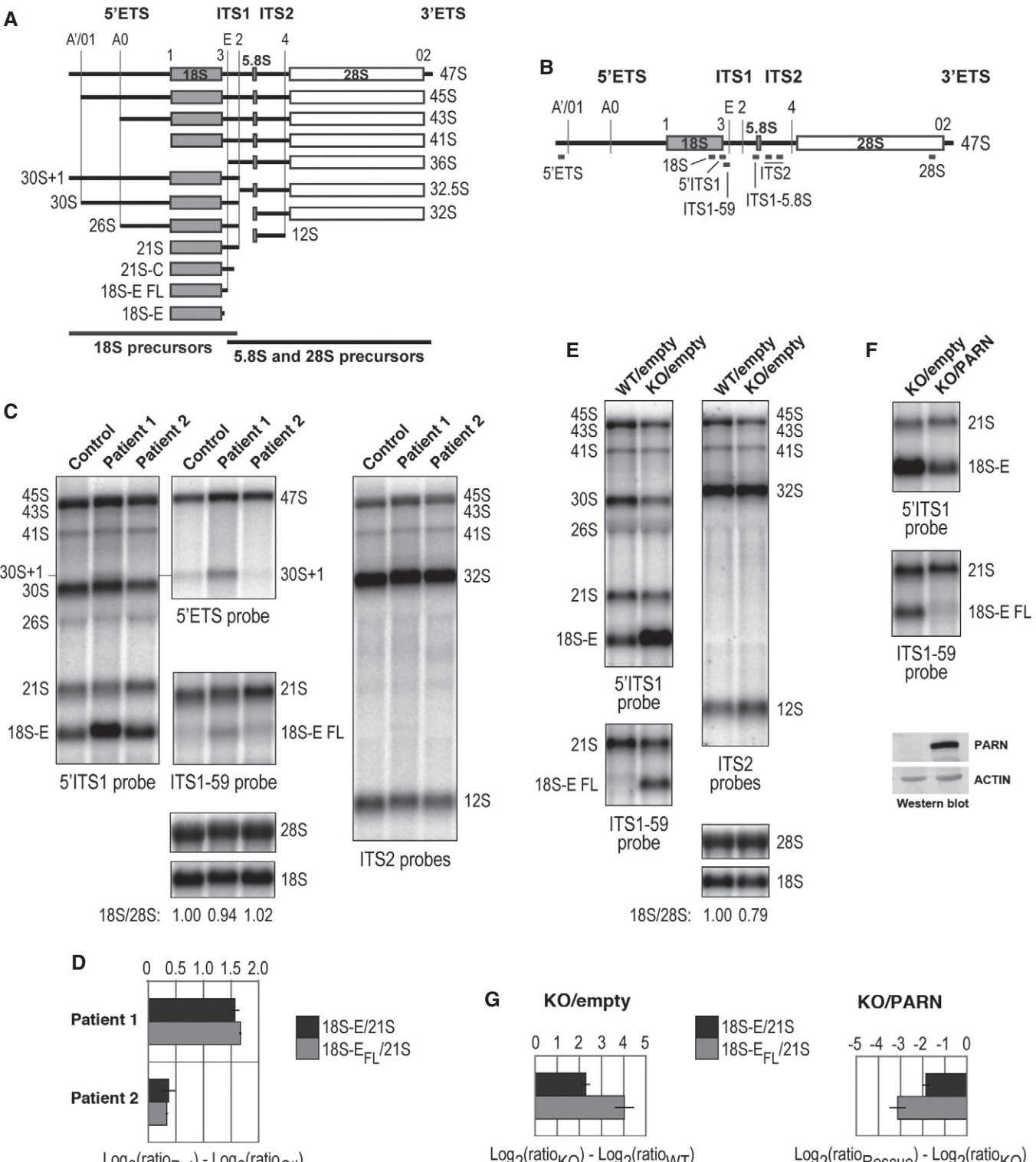

**Figure 6.**

Figure 6. PARN dysfunction impacts human pre-ribosomal RNA processing.

A Schematic representation of human ribosomal precursors with rRNAs sequences displayed as gray (18S, small ribosomal subunit) or white boxes (5.8S and 28S rRNAs, large ribosomal subunit). These three rRNAs are flanked by external (5′ETS, 3′ETS) and internal transcribed spacers (ITS1, ITS2). The position of endonucleolytic cleavages is represented by vertical lines along these transcribed spacers or corresponds to 5′ and 3′ ends of rRNA sequences.

B Graphical representation of DNA probes used for Northern blot hybridization displayed along the 47S primary rRNA transcript.

C Northern blot analysis of pre-rRNAs from control and patient B-LCLs.

D $Log_2$ values of 18S-E/21S and 18S-$E_{FL}$/21S for P1 and P2 were normalized to the values of the control and presented in a graphical format, with Log averages $\pm$ standard deviation for two independent experiments.

E Northern blot analysis of pre-rRNAs from HT1080[PARN WT] and HT1080[PARN KO] cells, transduced with an empty vector.

F Ectopic expression of PARN was induced in HT1080[PARN KO] cells and compared to cells transduced with an empty vector. Western blot relative to actin assessed PARN levels.

G Quantitative analyses of 18S-E/21S and 18S-E FL/21S ratios were as described in (D) for HT1080[PARN KO] cells relative to HT1080[PARN WT] (left panel; see (E)) and for HT1080[PARN KO] cells rescued by ectopic expression of PARN relative to HT1080[PARN KO] cells transduced with an empty expression vector (right panel; see (F)), with Log averages $\pm$ standard deviation for four independent experiments.

## Discussion

Disease resulting from biallelic *PARN* mutations has been reported in only 9 patients so far (Dhanraj *et al*, 2015; Moon *et al*, 2015; Tummala *et al*, 2015; Burris *et al*, 2016). We here describe two unrelated patients carrying novel biallelic *PARN* mutations and exhibiting a phenotype corresponding to Høyeraal–Hreidarsson syndrome. PARN, via its control of h*TR* maturation and stabilization, participates in telomerase activity and consequently in telomere maintenance (Moon *et al*, 2015). However, the clinical severity of PARN-deficient patients, including ours, suggests that, besides h*TR* down-regulation, other important biological processes may be impaired. In this study, we generated a HT1080[PARN KO] cell line carrying an inducible complementing PARN allele to carefully examine the consequences of PARN depletion in human cells. We confirmed the reduced h*TR* expression and its mislocalization to the cytoplasm of PARN KO cells (Shukla *et al*, 2016) and found a down-regulation of *TRF1, TRF2, POT1, TPP1,* and *DKC1* mRNA levels. We also observed an induction of *p53* mRNA and protein levels by about twofold.

Importantly, we found that PARN-depleted cells exhibit multiple telomeric defects reminiscent to those reported in shelterin-deficient cells (Sfeir & de Lange, 2012). Most telomeric aberrations were rescued after 72 h of PARN induction, together with a complete rescue of *TRF1, TRF2, POT1,* and *TPP1* mRNA levels. Since the 72-h treatment with doxycycline was not sufficient to promote telomerase-dependent re-elongation of telomeres, our study suggests that PARN is able to promote telomere stability independently of telomere length, likely by regulating shelterin expression. On the other hand, complementation by PARN did not rescue *DKC1* mRNA levels in PARN KO cells. Our data indicated that the down-regulation of *DKC1* mRNA levels in PARN KO cells was dependent on p53 induction in these cells (Fig 8), in agreement with the previous observation that the hyperactive p53[A31] allele down-regulates DKC1 expression in mouse cells (Simeonova *et al*, 2013). Supporting a role for up-regulated p53 in the down-regulation of *DKC1* mRNA in PARN-depleted cells, we found that p53 expression levels were not back to normal upon complementation with PARN. In light of these results, we conclude that the impact of PARN depletion on *TRF1, TRF2, POT1,* and *TPP1* mRNA levels is independent on p53 induction (Fig 8). Similar to what was recently reported for *p53* mRNA (Shukla *et al*, 2019), PARN deadenylase activity may stabilize the mRNAs of these telomere-related genes through the up-regulation

of some miRNAs. Alternatively, PARN may directly stabilize *TRF1, TRF2, POT1,* and *TPP1* mRNAs by deadenylating their 3′ ends. Additional experiments will be needed to understand how PARN modulates the mRNA levels of telomere-related genes.

The reason why PARN complementation failed to rescue *p53* mRNA levels in our experimental cellular system is currently unknown but fits with the previous hypothesis that depletion of PARN may induce compensatory cellular mechanisms through the up-regulation of other deadenylases, including the Ccr4a, b, or d enzymes (Zhang & Yan, 2015). Along this line, we detected an up-regulation of h*Ccr4d* mRNA in the PARN KO cells (data not shown). Our results also provide evidence that PARN does not directly regulate global TERRA biogenesis or degradation, suggesting that the negative impact of PARN depletion on telomere stability is not due to deregulated TERRA levels.

We further showed that both PARN-deficient patient cells and PARN KO cells exhibited defective rRNA biogenesis that was reverted by PARN complementation, thus establishing that defective rRNA biogenesis in a PARN-deficient context is not due to p53 dysregulation.

Altogether, the experimental cellular system that we developed to study the multiple consequences of PARN depletion revealed an important role for the enzyme in telomere elongation, telomere stability, and rRNA biogenesis, and clarified the impact of the associated p53 up-regulation, previously proposed to be central in the cellular phenotypes of telomere-related mutations (Mason & Bessler, 2015). Our data indicate that, in the context of PARN depletion, p53 up-regulation may only down-regulate a limited number of telomere-related genes and is not the cause of telomere instability or impaired rRNA biogenesis (Fig 8). Further supporting our results with the experimental human cellular system was the observation that the early embryonic lethality of the PARN KO mice we developed by CRISPR/Cas9 was not compensated by p53 depletion. Because $mTR^{-/-}$ and $mTERT^{-/-}$ mice are viable (Blasco *et al*, 1997; Liu *et al*, 2000), these data are consistent with Parn having essential roles besides its function in telomerase activity. Accordingly, we confirmed that mouse PARN participates in rRNA biogenesis.

Altogether, our study strongly supports the involvement of PARN enzyme in multiple important biological pathways, in both mouse and humans, which, together, likely contribute to the pathophysiology of the disease (Fig 8). The lethality of Parn KO mice suggests that the severe PARN mutations identified in patients are either hypomorphic or lead to the acquisition of compensatory mechanisms through, for instance, the up-regulation of other

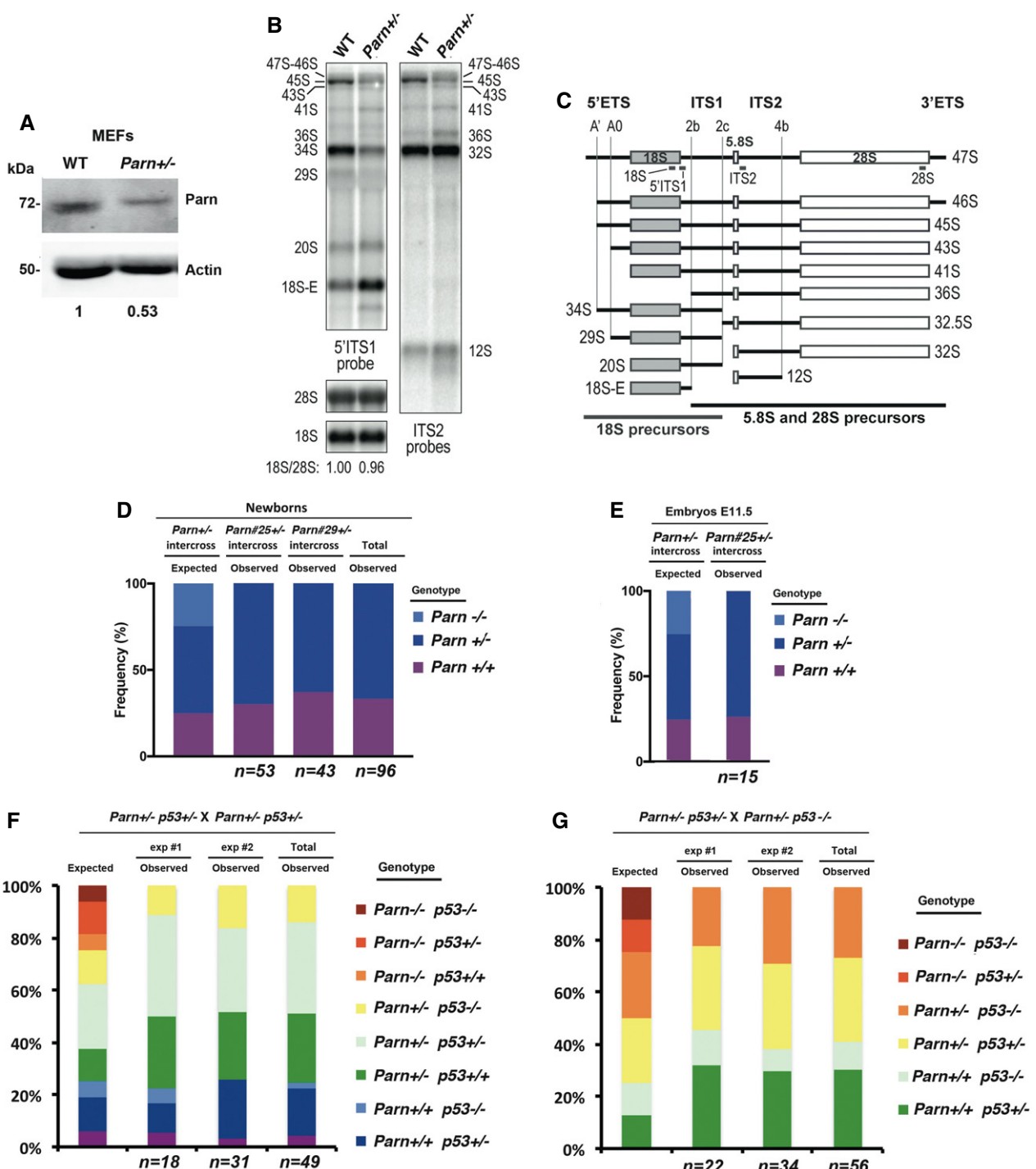

**Figure 7. The early embryonic lethality of *Parn* KO is not rescued in *p53⁻/⁻* mice.**

A  Western blot analysis of Parn expression in control and *Parn⁺/⁻* MEFs.

B  Northern blot analysis of total RNAs from WT and *Parn⁺/⁻* MEFs.

C  Schematic representation of mouse rRNA precursors and position of DNA probes used in Northern blot.

D  Frequencies of expected and observed newborns of the indicated genotype obtained from *Parn⁺/⁻* intercrosses. *n* indicates the number of animals analyzed. Statistical significance (*P* = 1.5417E-08) of the differences between the observed and expected genotype distributions was assessed by chi-square test.

E  Frequencies of expected and observed E11.5 embryos of the indicated genotype obtained from *Parn⁺/⁻* intercrosses. *n* indicates the number of animals analyzed. Statistical significance (*P* = 0.02535) of the differences between the observed and expected genotype distributions was assessed by chi-square test.

F, G  p53 knock-out does not rescue *Parn⁻/⁻*. No *Parn⁻/⁻ p53⁻/⁻* mouse was born from various combinations of *Parn⁺/⁻* and either *p53⁺/⁻* or *p53⁻/⁻* crosses (49 and 56 pups, respectively). Statistical significance (*P* = 0.000886) of the differences between the observed and expected genotype distributions was assessed by chi-square test.

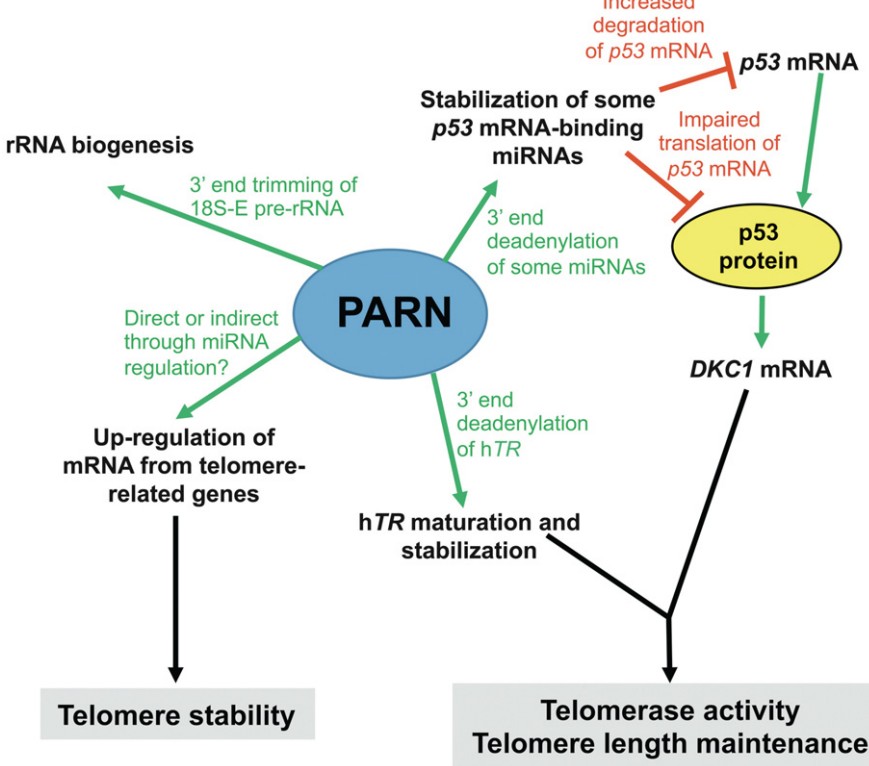

**Figure 8. The multiple substrates of PARN in telomere biology and rRNA biogenesis.**

PARN regulates the expression of telomere-related gene transcripts, including shelterin transcripts and the h*TR* RNA component of telomerase. How PARN stabilizes h*TR* has been well described and involves its 3′-end deadenylation. However, at present, the mechanisms underlying PARN-dependent regulation of telomere-related gene transcripts are still unknown and may involve direct or indirect regulations through 3′-end deadenylation of either the shelterin mRNAs or some miRNAs that bind to these mRNAs. Our data also show that PARN regulates the level of *DKC1* mRNA through the control of p53 levels. Taken together, PARN impacts on both telomere stability and telomere length. An additional role of PARN is to regulate rRNA maturation in both human and mouse cells. Our data suggest that PARN-dependent regulation of rRNA maturation occurs independently of p53. The involvement of PARN in various aspects of cell biology likely explains the severity of the phenotype in Høyeraal–Hreidarsson syndrome patients carrying biallelic *PARN* mutations.

deadenylases. Heterozygous *PARN* mutations have been associated with idiopathic pulmonary fibrosis and rheumatoid arthritis-interstitial lung disease associated with short telomeres (Stuart *et al*, 2015; Juge *et al*, 2017). Our experiments with $Parn^{+/-}$ MEFs revealed that PARN amount is limiting in cells. A careful follow-up of parents carrying heterozygous PARN mutation is therefore needed to anticipate potential development of pulmonary fibrosis, progressive bone marrow failure, or other ailments associated with premature aging.

# Materials and Methods

### Study approval

Informed and written consent was obtained from donors, patients, and families of patients. The study and protocols comply with the 1975 Declaration of Helsinki as well as with the local legislation and ethical guidelines from the Comité de Protection des Personnes de l'Ile de France II and the French advisory committee on data processing in medical research. A consent was obtained from the parents of P1 to publish the patient's photographs.

### Cells

Control fibroblasts were obtained from skin biopsies from pediatric healthy donors (3 years of age). Control fibroblasts were indifferently from healthy male or female donors (no difference in phenotype was noticed with gender). Fibroblasts were transformed by the large T antigen from SV40T as previously described (Buck *et al*, 2006). The HT1080 human fibrosarcoma cell line was kindly provided by Arturo Londoño-Vallejo (Institut Curie, Paris, France). All cell lines were checked for mycoplasma contamination.

### Whole exome sequencing

Exome capture was performed using the SureSelect Human All Exon Kit (Agilent Technologies®, Santa Clara, CA). Agilent SureSelect Human All Exon (54 Mb, Clinical research Exome) libraries were prepared from 3 μg of genomic DNA sheared with an Ultrasonicator (Covaris®, Woburn, MA) as recommended by the manufacturer. Barcoded exome libraries were pooled and sequenced using a HiSeq2500 (Illumina®, San Diego, CA) generating 130 × 130 paired-end reads. After demultiplexing, sequences were mapped on the human genome reference (NCBI build37/hg19 version) with BWA.

The mean depth of coverage obtained from the exome library was 138X with > 99% of the targeted exonic bases covered by at least 15 independent reads and > 97% by at least 30 independent sequencing reads (> 99% at 15× and > 97% at 30×). Variant calling was carried out with the Genome Analysis Toolkit (GATK), SAMtools, and Picard Tools. Single nucleotide variants were called with GATK Unified Genotyper, whereas indel calls were made with the GATK IndelGenotyper_v2. All variants with a read coverage ≤ 2× and a Phred-scaled quality of ≤ 20 were filtered out. All the variants were annotated and filtered using Polyweb, an in-house developed annotation software.

## Constructs and cDNA analysis

Total RNA from patient and control fibroblasts or B-LCL was extracted using TRIzol reagent (Invitrogen, Grand Island, NY) according to the manufacturer's instructions. Reverse transcription was performed using a SuperScript First-Strand Synthesis Kit (Invitrogen). Nucleotide numbering reflects cDNA numbering with +1 corresponding to the A of the ATG translation initiation codon in the reference sequence. The initiation codon is codon 1. PARN ORF was PCR-amplified from cDNA and cloned into an inducible pCW57-lentiviral vector by replacing the GFP cassette for complementation experiments (Addgene Plasmid #71783).

## Targeted resequencing by NGS (capture by hybridization approach) to detect copy number variations

Illumina compatible barcoded genomic DNA libraries were constructed according to the manufacturer's instructions (Ovation Ultralow, NuGen Technologies). Briefly, 1–3 μg of each patient's genomic DNA was mechanically fragmented to a median size of 200 bp using a Covaris sonicator. 100 ng of fragmented dsDNA was end-repaired, and adaptors containing a specific 8 bases barcode were ligated to the repaired ends. DNA fragments were then PCR-amplified to get the final pre-capture barcoded libraries that were pooled at equimolar concentrations (a pool of 15 libraries was prepared). The capture process was performed using SureSelect reagents (Agilent), 750 ng of the pool of pre-capture libraries, and home-made biotinylated probes. The biotinylated ssDNA probes were designed and prepared to cover a 194-kb chromosomal region including the complete *PARN* gene on chromosome 16. During the capture process, barcoded library molecules complementary to the biotinylated beads were retained using streptavidin-coated magnetic beads and PCR-amplified to generate a final pool of post-capture libraries covering the targeted chromosomal region on chromosome 16. In total, a pool of 7 libraries (6 samples and 1 DNA control), covering a 194-kb territory including the entire *PARN* gene, was sequenced on an Illumina HiSeq2500 (paired-end sequencing, 130 × 130 bases, high-throughput mode, 7 samples on half of a FlowCell lane). After demultiplexing, sequences were aligned to the reference human genome hg19 using the Burrows-Wheeler Aligner (Li & Durbin, 2010). The mean depth of coverage obtained per sample was ≥ 600× to enable more accurate copy number variation analysis. Downstream processing was carried out with the Genome Analysis Toolkit (GATK), SAMtools, and Picard, following documented best practices (http://www.broadinstitute.org/gatk/guide/topic?name = best-practices). Variant calls were made with the GATK Unified Genotyper. The annotation process was based on the latest release of the Ensembl database. Variants were

annotated and analyzed and prioritized using the Polyweb/PolyDiag software interface designed by the Bioinformatics platform of University Paris Descartes.

## Telomere restriction fragment analysis

DNA (800 ng) was digested with *Hinf*I and *Rsa*I enzymes, resolved by a 0.7% agarose gel, and transferred to a nylon membrane. Hybridization was performed using EasyHyb solution (Roche) and $\gamma$-$^{32}$P-labeled $(TTAGGG)_4$ probe. After washes, membranes were exposed over a PhosphorImager (AGFA). PhosphorImager exposures of telomere-probed Southern blots were analyzed with the ImageJ program. The digitalized signal data were then transferred to Microsoft Excel and served as the basis for calculating mean TRF length using the formula L = (ODi)/(ODi/Li), where ODi = integrated signal intensity at position i, and Li = length of DNA fragment in position i.

## FISH and Q-FISH

Seeded cells were arrested in metaphase with 60 ng/ml colcemid (KaryoMAX, Invitrogen) for 30 min, harvested, and resuspended in 75 mM KCl for 15 min at 37°C. Cells were then fixed in 3:1 methanol/acetic acid and dropped onto glass slides. Metaphase spreads were fixed in 4% formaldehyde in PBS for 2 min and dehydrated with sequential immersions into 50, 70, and 100% ethanol baths for 2 min each and then air-dried. Telomere PNA-FISH was performed in 70% deionized formamide, 1% blocking reagent (Roche), and 0.3 μg/ml Cy3-$(C_3TA_2)_3$ PNA probe (Panagene). DNA was denatured for 5 min at 80°C, then hybridized for 2 h at room temperature (RT). Slides were next washed as follows: 2 × 15 min at RT in 70% formamide, 10 mM Tris pH 7.2 and 3 × 5 min in 50 mM Tris pH 7.5, 150 mM NaCl, and 0.05% Tween-20. Slides were then dehydrated in ethanol, air-dried, and counterstained with DAPI mounted in Vectashield (Vector Laboratories) to estimate total telomere fluorescence intensity in FISH experiments (Q-FISH). National Institutes of Health software (ImageJ) was used for the quantitative analysis of images. Telomere intensity in samples was normalized to fluorescence intensity of telomeric probe labeling telomeres from Muntjac cells seeded on the same coverslip.

## Detection of TIF

For telomere dysfunction-induced foci (TIF) analysis, cells grown on coverslips were fixed with 2% paraformaldehyde for 10 min, permeabilized with 0.1% Triton X-100 for 30 min, and incubated with anti-53BP1 (22760, Santa Cruz; 1:200) for 1h at RT. After washing and incubation with the secondary antibody, cells were washed in PBS and dehydrated in sequential ethanol baths, and FISH was performed as described above.

## Senescence-associated β-galactosidase staining

Primary fibroblasts were fixed at room temperature for 10 min in 4% paraformaldehyde in PBS, washed in PBS, and then stained in β-galactosidase fixative solution (Senescence β-Galactosidase Staining Kit, # 9860, Cell Signaling) at 37°C for 16 h before cell imaging.

## Western blotting

Cells were lysed for 20 min on ice in lysis buffer containing 50 mM Tris (pH 8.0), 2 mM EDTA, 1% Triton X-100, 1% phosphatase inhibitor cocktails (Sigma), and protease inhibitors (Roche). After centrifugation, the supernatant was harvested and protein concentration was quantified with the Bradford assay. After SDS–PAGE, proteins were transferred to PVDF Immobilon-P membrane (Millipore). Then, the membrane was incubated for 1h in Odyssey blocking buffer (TBS), followed by an incubation with anti-PARN antibody (Abcam, ab188333, 1:1,000), then washed and incubated with goat anti-rabbit secondary antibody (Li-Cor IRDye 800CW Infrared Dye, 1:15,000 dilution). The presence of PARN protein was detected by infrared fluorescence according to the manufacturer's protocol (Odyssey CLx Imaging System). The blot was then incubated with anti-actin or anti-GAPDH (Sigma) as loading control. For the WB of Fig 4I, 30 μg of protein extracts was processed as described previously (Arnoult *et al*, 2012) using the following antibodies: anti-TRF2 (Novus Biologicals, NB110-57130, 1:20,000), anti-p53 (Santa Cruz, sc-126, 1:10,000), and anti-β-actin (Sigma, A5441, 1:100,000). Secondary antibodies were as follows: anti-rabbit-HRP (Enzo Life Sciences, ADI-SAB-300-J, 1:10,000) and anti-mouse-HRP (Abcam, ab205719, 1:2,000). SuperSignal West Pico Chemiluminescent Substrate reagent (Thermo Scientific, #34580) was used for revelation, and signals were quantified on films using ImageJ software.

## h*TR* FISH

The protocol described in Stern *et al* (2012) was followed using 12.5 ng of the following FISH probes (Eurofins):

hTR#1: Cy5-GCTGACATTTTTTGTTTGCTCTAGAATGAACGGTGGA AGGCGGCAGGCCGAGGCTT
hTR#4: Cy5-CTCCGTTCCTCTTCCTGCGGCCTGAAAGGCCTGAACCT CGCCCTCGCCCCCGAGAG
hTR#5: Cy5-ATGTGTGAGCCGAGTCCTGGGTGCACGTCCCACAGCT CAGGGAATCGCGCCGCGCGC

## siRNA transfection

Transfections with siRNAs were performed as described previously (Arnoult *et al*, 2012) using the following siRNAs (Eurogentec): sip53 (Bergamaschi *et al*, 2003), 5′-CUACUUCCUGAAAACAACG, and siLuci (Arnoult *et al*, 2012), 5′-CUUACGCUGAGUACUUCGA. Cells were collected 72 h after transfection.

## qRT–PCR

Total RNA was isolated using TriPure reagent (Sigma), and qRT–PCRs were performed as described previously (Arnoult *et al*, 2012) using the primers listed in Appendix Table S2.

## Analysis of pre-ribosomal RNA processing

Total RNA isolated from human and mouse cell lines by using TRIzol was quantified with a NanoDrop spectrophotometer. Samples corresponding to 3 μg total RNAs were separated on a 1.1% agarose gel containing 1.2% formaldehyde and 1× Tri/Tri buffer (30 mM triethanolamine, 30 mM tricine, pH 7.9), transferred to Hybond N$^+$ nylon membrane (GE Healthcare), and cross-linked under UV light. After incubation in hybridization buffer (6× SSC, 5× Denhardt's solution, 0.5% SDS, 0.9 μg/ml tRNA), the 5′-radiolabeled oligonucleotide probe was added and incubated overnight (45–55°C). The membrane was washed and exposed to a PhosphorImager screen, which was revealed using a Typhoon Trio PhosphorImager (GE Healthcare) and quantified using the MultiGauge software. The human probes used were as follows:5′ETS (5′-AGACGAGAACGCCTGACACGCACGGCAC-3′), 5′ITS1 (5′-CCTCGCCCTCCGGGCTCCGTTAATGATC-3′), ITS1-59 (5′-GCGGTGGGGGGGTGGGTGTG-3′), and a mixture of ITS2-1 (5′-CT GCGAGGGAACCCCCAGCCGCGCA-3′) and ITS2-2 (5′-GCGCGACGGC GGACGACACCGCGGCGTC-3′). The mouse probes were as follows: 5′ITS1 (5′-GCTCCTCCACAGTCTCCCGTTAATGATC-3′) and ITS2 (5′-ACCCACCGCAGCGGGTGACGCGATTGATCG-3′). The same probes were used to hybridize human and mouse 18S (5′-TTTACTTCCTCTA GATAGTCAAGTTCGACC-3′), and 28S rRNAs (5′-CCCGTTCCCTTGG CTGTGGTTTCGCTAGATA-3′).

## *In vitro* deadenylation assay

Deadenylation assay was performed as described previously by Tummala *et al* (2015). Whole-cell extracts were incubated for 1 h at 30°C in deadenylation buffer (20 mM Tris–HCl pH 7.9, 50 mM NaCl, 2 mM MgCl$_2$, 10% glycerol, 1 mM β-mercaptoethanol) with fluorescein-5′ labeled 16-mer RNA oligonucleotide (5′-CCUUUCC AAAAAAAA-3′). Samples were then heated at 85°C for 3 min in RNA loading buffer and run in a denaturing PAGE using 20% acrylamide:bisacrylamide (19:1) and 50% urea. Results were analyzed by phosphorimager fluorescent image analyzer FLA-3000 (Fuji-Film).

## Telomerase activity assays

Cells were lysed in 1× CHAPS lysis buffer (TRAPeze S7700, Millipore), and proteins were quantified by Bradford assay (Bio-Rad). Four dilutions of protein extracts (500, 250, 125, and 62.5 ng) were assayed for TRAP. TRAP products were separated on TBE/acrylamide:bisacrylamide (19:1) gel and visualized by staining with SYBR Gold Nucleic Acid Gel Stain (Invitrogen).

## CRISPR/Cas9 gene inactivation in HT1080 cells

The 5′-CCGACTTCTTCGCCATCGAT-3′ located in PARN exon 2 was used as gRNA and cloned into the pX330-U6-Chimeric_BB-CBhhSpCas9 plasmid (a gift from F. Zhang, the Broad Institute of Massachusetts Institute of Technology and Harvard University, Cambridge, MA; plasmid 42230; Addgene; Ran *et al*, 2013) for transfection into HT1080 cells. Cells were then cloned and assessed by Sanger sequencing for the presence of *PARN* mutations.

## Generation of *Parn* KO mouse model

The generation of Parn KO mice was done in the Mouse Genetics Engineering Center from Institut Pasteur (Paris, France). All procedures were reviewed and approved by the Ethics Committee of Institut Pasteur CETEA (2013-0136). All efforts were made to

minimize animal suffering and to reduce the number of animals required for the experiments. A guide RNA sequence was selected on exon 4 of the murine *Parn* gene using the CRISPOR Web tool (http://crispor.tefor.net/crispor.py; Haeussler *et al*, 2016). Double-stranded DNA oligonucleotides corresponding to the selected guide RNA were cloned into the pX330-U6-Chimeric_BB-CBh-hSpCas9 vector (generous gift from Feng Zhang, Addgene #42230) according to F. Zhang Lab's recommendations (Cong *et al*, 2013). For mutagenesis scoring and mouse genotyping, genomic DNA surrounding the guide RNA target sequences was PCR-amplified (forward: 5′-tctggagttgactagtgtcc-3′; reverse: 5′-ttcatgctgactgactctgg-3′) and the resulting PCR products were Sanger-sequenced. To generate mutant mouse lines, zygotes were microinjected with pX330 according to the protocol of Mashiko *et al* (2013). F0 mice were obtained, and tail DNA was analyzed by Sanger sequencing after PCR amplification of the *Parn*-targeted locus. F0 mice were crossed onto C57BL6/J to segregate the CRISPR/Cas9-generated mutant alleles. Two F0-derived F1 mice (#25 and #29) were selected. CRISPR/Cas9 mutagenesis resulted in a 2 base-pair deletion in #25 and a 1 base-pair deletion in #29 and subsequent frameshift in both cases (Appendix Fig S6). Each line was then backcrossed up to 7 times on C57BL6/J (B6) to segregate away any off-target event outside of chromosome 16. PCR amplification and direct sequencing of the eleven genomic loci predicted by the CRISPOR Web tool to be putatively off-targeted did not reveal any mutation in $Parn^{+/-}$ heterozygous animals (Appendix Table S1).

### Mice

The $p53^{-/-}$ mice were described previously (Vera *et al*, 2013) and are on a mixed B6/129 background and maintained by backcrosses onto the C57BL6/J strain. All mice were housed and handled in TAAM-CNRS-UPR44-Orléans animal facility (Agreement number n°D45-234-6, CNRS, France). Mice were kept in a temperature-controlled environment with a 12-h/12-h light–dark cycle, with a standard diet and water *ad libitum*. All animals were treated in accordance with the Guide for the Care and Use of Laboratory Animals as adopted by INSERM with full respect to the EU Directive 2010/63/EU for animal experimentation. After genotyping, all the mice were euthanized. All the experiments were performed under appropriate license from the local ethical committee and the French Ministry of Education and Research (#01501.03).

### Statistical analyses

Various Web tools and Microsoft Excel were used for statistical analyses. When needed, the Shapiro–Wilk test was applied to check for normal distribution of the data (http://www.sthda.com/french/rsthda/shapiro-wilk.php). For the graph of Fig 2D, normal distribution was not observed ($P < 0.05$). The Bartlett test for comparison of variances was applied that similarly led to exclude the hypothesis of a similarity of variance between the groups. We thus applied the Kruskal–Wallis non-parametric test for data of Fig 2D. For Figs 2I and J, and 4A, a chi-square test was applied that does not require any Shapiro–Wilk test. Similarity of the variance between the compared groups was assessed

by a Bartlett test (https://biostatgv.sentiweb.fr/?module=tests/anova). When normal distribution was observed, the Student's *t*-test was used to compare the differences between means using Excel. When *P* values provided by the Bartlett test were lower than 0.05, the correction for unequal variance was applied for Student's *t*-tests. In the absence of normal distribution of the data, the non-parametric Kruskal–Wallis test was used (http://astatsa.com/KruskalWallisTest/). The chi-square test was also applied in some cases where the number of observations was really high (above 1,400) using https://www.socscistatistics.com/tests/chisquare/default2.aspx and Excel. Most graphs indicate s.e.m. (standard error of the mean), and the number of replicates is provided in the figure legends.

For the mouse study, by using G*Power tool, we established that, to demonstrate a lethality with crossing of hetXhet animals, with a power of 0.8 and a *P* value < 0.05, 11 animals were required. We obtained in total 96 animals without homo mice, demonstrating the lethality. The same holds true with embryos (15 animals analyzed).

## Data availability

Sequence data have been deposited at the European Genome-phenome Archive (EGA), which is hosted by the EBI and the CRG, under accession number EGAS00001003623.

**Expanded View** for this article is available online.

### Acknowledgements
We thank the patients and their families for their contribution. We thank Jerry W. Shay and Woodring E. Wright for the kind gift of the Muntjac cells. P.R. is grateful to Prof. Alain Fischer for discussions, advice, and support. We acknowledge the excellent technical assistance of Alicia Fernandez for generating B-LCL (CRB, Imagine Institute, Paris, France) and Nikenza Vicecon for the h*TR* FISH. This work has been supported by institutional grants from INSERM, Ligue Nationale contre le Cancer (Equipe Labellisée La Ligue), Institut National du Cancer INCa, GIS-Institut des maladies rares, and FNRS (Fonds National de la Recherche Scientifique, Belgium). P.E.G. and M.F.O. are supported by Agence Nationale de la Recherche (ANR 2015 AAP générique CE12-0001-DBA Multigenes), and the EuroDBA project is funded by the ERA-NET program E-RAR3 (ANR-15-RAR3-0007-04). P.R. and M.F.O. are scientists from Centre National de la Recherche Scientifique (CNRS). H.E. and F.P. were supported by grants from the Télévie/FNRS and FRIA/FNRS. A.D. is a scientist from the FNRS.

### Author contributions
PR initiated the project and performed whole exome sequencing analysis. PR and AD supervised and coordinated the project. PF, FBB, and MP identified affected patients. MB, LK, and PR performed Sanger sequencing and cloning. MB, HE, LK, FP, AD, and PR generated and performed phenotypic analyses of the PARN KO and PARN-deficient cells from patients. M-FO' and P-EG performed rRNA biogenesis studies. PR and F L-V conceived and generated the PARN KO mouse model. IC performed structural analysis. AD, HE, and FP performed qRT–PCR analyses of TERRA and telomere-related gene transcripts (with WB) and sip53 experiments. h*TR* FISH experiment was performed in AD's laboratory. PR and AD prepared figures and wrote the manuscript, supported by PF, M-FO', P-EG, IC, and J-PV.

**The paper explained**

**Problem**

PARN deficiency causes Høyeraal–Hreidarsson (HH) syndrome, a rare and severe telomere biology disorder. It has been proposed that most of the clinical features found in PARN-deficient patients result from telomere length defect. The up-regulation of p53 was also proposed to contribute to the phenotype. To refine the functional consequences of PARN deficiency, we analyzed PARN-mutated cells from two unrelated HH patients carrying novel biallelic PARN mutations and a human PARN KO cell line with inducible PARN complementation. Furthermore, we generated a Parn KO mice.

**Results**

Our analysis demonstrated that PARN-deficient cells exhibit not only reduced telomere length but also increased instability, independently of telomere length. Furthermore, PARN defect reduces the steady-state mRNA levels of a set of genes involved in telomere stability and impairs rRNA biogenesis. Apart from the impact on DKC1 mRNA levels, these effects are independent from p53 up-regulation in PARN-deficient cells. Lastly, we demonstrated that Parn KO mice are embryonic lethal at a very early stage and that the lethality is not reverted by p53 KO.

**Impact**

PARN deficiency does not only cause telomere length defect, but also cause telomere instability and impaired rRNA biogenesis. The many functional consequences of PARN deficiency might explain, at least in part, the clinical severity found in HH patients carrying biallelic PARN mutations.

## Conflict of interest

The authors declare that they have no conflict of interest.

## For more information

(i) https://gnomad.broadinstitute.org/gene/ENSG00000140694

(ii) https://www.omim.org/entry/604212

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
