## [Review Process File · EMBO Molecular Medicine]

Impaired telomere integrity and rRNA biogenesis in PARN-deficient patients and knock-out models

Maname Benyelles, Harikleia Episkopou, Marie-Françoise O'Donohue, Laëtitia Kermasson, Pierre Frange, Florian Poulain, Fatma Burcu Belen, Meltem Polat, Christine Bole-Feysot, Francina Langa-Vives, Pierre-Emmanuel Gleizes, Jean-Pierre de Villartay, Isabelle Callebaut, Anabelle Decottignies, Patrick Revy

Review timeline:

Submission date:	13 December 2018
Editorial Decision:	7 February 2019
Revision received:	28 February 2019
Editorial Decision:	10 April 2019
Revision received:	24 April 2019
Accepted:	9 May 2019

Editor: Céline Carret

Transaction Report:

1st Editorial Decision

7 February 2019

Thank you for the submission of your manuscript to EMBO Molecular Medicine. We have now heard back from the three referees whom we asked to evaluate your manuscript.

You will see from the comments pasted below that Referees #1 and #2 are supportive of publication while Referee #3 raises an important concern as novelty being compromised. Referee #2 also regrets that the paper only provides weak mechanistic insights and this aspect of the study should be improved. As we do like the translational potentials highlighted in your work, we would like to encourage you to address the mechanistic issues, knowing that this would increase the novelty of the findings. In addition, we would like to ask you to thoroughly discuss the papers commented by Referee #3.

We would therefore welcome the submission of a revised version within three months for further consideration and would like to encourage you to address all the criticisms raised as suggested to improve conclusiveness and clarity. Please note that EMBO Molecular Medicine strongly supports a single round of revision and that, as acceptance or rejection of the manuscript will depend on another round of review, your responses should be as complete as possible.

I look forward to receiving your revised manuscript.

***** Reviewer's comments *****

Referee #1 (Remarks for Author):

Benyelles et al. describe the pleiotropic consequences of bilallelic pathogenic PARN mutations in telomere metabolism and rRNA biogenesis in two unrelated patients with Høyeraal-Hreidarsson syndrome. They found that, in addition to critical telomere shortening, patients' cells also display down regulation of a series of sheltering proteins (TRF1, TRF2, TPP1, RAP1, and POT1), which is rescued in PARN-deficient cell lines by inducible PARN expression. They also show that ribosomal RNA biogenesis in human fibroblasts from patients and heterozygous Parn KO mice. Experiments are consistent and well conducted and conclusions are well supported by data. I have only minor comments.

- 1) line 69: Høyeraal-Hreidarsson syndrome is not the most severe clinical variant of telomeropathies, but rather Revesz syndrome (Alter et al. 2012)
- 2) Gene names should be italicized.
- 3) line 110: Bone marrow aspirate does not demonstrate pancytopenia. Pancytopenia is diagnosed in peripheral blood.
- 4) line: line 120: likewise, marrow aspirate does not demonstrate marrow cellularity, which is usually assessed by BM biopsy.
- 5) line 166: It is interesting that patients did not present somatic TERT promoter variants. It has recently been shown that these variants are restricted to patients with either TERT or TERC germline mutations (Gutierrez-Rodriguez, 2018).
- 6) What is the explanation of hTR mislocalization to the cytoplasm? Is it the result of abnormal poly(A) tails? In these instances, is hTR detached from hTERT?

Referee #2 (Comments on Novelty/Model System for Author):

The technical quality of this work is strong based on the visually apparent quality of gels, magnitude of biological effects (both absolute and relative), and the extent of quantitation and statistical tests.

The novelty of this work is good because the study of PARN is still a relatively new topic in the telomere field, and the progress made by the authors that is reported in this manuscript is substantial.

The medical impact is important given this is major mechanistic progress on the study of "telomeropathies" that are rare but often severe. Such diseases and disease mechanisms are an emerging field that spans multiple clinical fields.

The model system is patient samples and the mouse, and the results are clearly synergistic and complementary.

Referee #2 (Remarks for Author):

This work describes substantial new genetic insights into the role of PARN in the severe human telomeropathy, Hoyeraal-Hreidarsson (HH) syndrome. The study is important given the pinpointing of the specific gene mutations causing diseases in two patients, who were evaluated extensively. The data quality is very high, and the experiments are well-conceived and executed. The work lays the foundation for future study to understand how PARN operates and underlies this disease in molecular-mechanistic terms (it does not advance the study of PARN mechanism substantially per se). Overall, this is a very solid, innovative research study, about which I became increasingly enthusiastic as I read the authors' manuscript.

That said, there is room for substantial improvement without extensive additional work. Foremost, the authors need to make it clearer what they believe their data add, if anything, regarding the mechanistic role of PARN in regulating hTR ncRNA and Shelterin-component mRNAs throughout the text (particularly by introducing this more extensively and then revisiting it in the Discussion and Abstract). If the authors do not conclude much about the mechanism(s) based on their data, then they alternatively need to explain in the Abstract and Discussion clearly that such questions as to how PARN is operating in regulating these RNAs remains largely unknown. As it stands presently, the key question of mechanism as to how PARN regulates these transcripts is left under-addressed, and this leaves the story, as written, more descriptive and less impactful than it could be if the authors commented further on this question as to PARN's functional activities.

Regardless of the degree to which the authors believe their data inform understanding of PARN mechanism, the Introduction also needs to inform the reader as to the current status of this question, beyond what is essentially just a single sentence (line 78 of page 3). Additionally, the authors should be clearer on line 83 of p. 4 as to what "modification" by PARN is being referred to: is it RNA nucleolytic cleavage or might this refer to some alternatively possible/known PARN enzymatic action beyond cleaving RNA?

Related to mechanism and functional relationships between PARN and its telomere-affecting targets, Figure 8 could be improved. Is it really necessary to show the same model twice; one for functional PARN and one for dysfunctional? I would think this could be illustrated more cogently, and perhaps (related to the above paragraphs) convey some information as to what specific mechanistic action by PARN is required for each of its promoting and inhibiting effects on the listed targets.

Finally, the authors do not really expand upon how novel the "pleiotropy" that they claim in the title, abstract, Discussion, etc. is for the field, nor precisely how they define it with respect to its targets and positive and negative roles that PARN has on them. The manuscript would be improved by the authors further expanding on why they decided this title was the most compelling by explaining if/why this is a new finding, and precisely in what respects.

Referee #3 (Remarks for Author):

The authors have identified two unrelated Hoyeraal-Hreidarsson individuals carrying novel biallelic PARN mutations. Using PARN-mutated cells from patients and a human PARN KO cell line with inducible PARN complementation, Benyelles and coworkers have found that PARP deficiency affects both the length and the stability of the telomere, reduces the mRNA levels of a subset of shelterins and of DKC1 and present aberrant ribosomal RNA biogenesis. The authors also generated a KO mice, which is embryonic lethal. The heterozygous mice also exhibit aberrant ribosomal RNA biogenesis.

Main concern

Although the different models (from patients, cells and mice) used in this work show robust and well-correlated results, the authors' findings are just the confirmation of previous published works. Thus, PARP deficiency in patients was reported to be accompanied by telomere defects several times already (Burris et al., 2016; Dhanraj et al., 2015; Moon et al., 2015; Tummala et al., 2015). Transient depletion of human PARN was also reported to be associated with a down-regulation of DKC1, RTEL1 and TRF1 transcripts (TUMmala et al., 2015) and, similarly, experiments in mouse myoblast reported a decrease abundance of Terf1, Terf2 and Rtel1 gene transcripts (Lee et al. 2012). Moreover, PARN-depleted cell lines present aberrant ribosomal RNA biogenesis (Ishikawa et al., 2017; Montellese et al., 2017). Because of all the above, I think the work from Benyelles and coworkers lacks of enough novelty to be published in EMBO Mol. Med.

***** Reviewer's comments *****

Referee #1 (Remarks for Author):

Benyelles et al. describe the pleiotropic consequences of bilallelic pathogenic PARN mutations in telomere metabolism and rRNA biogenesis in two unrelated patients with Høyeraal-Hreidarsson syndrome. They found that, in addition to critical telomere shortening, patients' cells also display down regulation of a series of sheltering proteins (TRF1, TRF2, TPP1, RAP1, and POT1), which is rescued in PARN-deficient cell lines by inducible PARN expression. They also show that ribosomal RNA biogenesis in human fibroblasts from patients and heterozygous Parn KO mice. Experiments are consistent and well conducted and conclusions are well supported by data. I have only minor comments.

We thank the reviewer for his/her enthusiastic appreciation of our manuscript and suggestions. We provide point-by-point responses below.

1) line 69: Høyeraal-Hreidarsson syndrome is not the most severe clinical variant of telomeropathies, but rather Revesz syndrome (Alter et al. 2012)

→ We agree with the comment that both HH and Revesz represent extreme severe variants of Dyskeratosis congenita. We replaced the sentence by (page 3): "HH syndrome and Revesz syndrome are rare disorders that represent the most severe clinical variants of DC (Alter, Rosenberg et al., 2012, Glousker et al., 2015)."

2) Gene names should be italicized.

→ We carefully verified that gene names, but not protein names, are all italicized.

3) line 110: Bone marrow aspirate does not demonstrate pancytopenia. Pancytopenia is diagnosed in peripheral blood.

→ We thank the reviewer for this remark. This mistake has been corrected accordingly.

4) line: line 120: likewise, marrow aspirate does not demonstrate marrow cellularity, which is usually assessed by BM biopsy.

→ Again, we thank the reviewer for pointing that out. This mistake has been corrected accordingly.

5) line 166: It is interesting that patients did not present somatic *TERT* promoter variants. It has recently been shown that these variants are restricted to patients with either *TERT* or *TERC* germline mutations (Gutierrez-Rodriguez, 2018).

→ Gutierrez-Rodriguez et al. indeed recently reported somatic *hTERT* promoter-activating mutations in patients harbouring germline *hTERT* or *hTERC* mutations. However, Maryoung et al. (2017) reported the presence of clones carrying somatic *hTERT* promoter-activating mutation in a patient with a heterozygous germline *PARN* mutation. Based on this observation, we tested whether somatic *hTERT* promoter-activating mutation could be detected in our patients but did not detect any. We added the reference "Gutierrez-Rodriguez et al. 2018" that was not included in our former manuscript.

6) What is the explanation of *hTR* mislocalization to the cytoplasm? Is it the result of abnormal poly(A) tails? In these instances, is *hTR* detached from *hTERT*?

→ According to the model proposed by Shukla et al (Nat Struct Mol Biol 2016), there is a competition between *hTR* assembly (H/ACA and snoRNP proteins), 3' end processing by PAPD5 poly(A) polymerase and PARN deadenylase and degradation by EXOSC10 (exosome) on one hand and cytoplasmic export and degradation by DCP2 and XRN1 on the other hand. Upon depletion of either DKC1 or PARN, *hTR* is destabilized and mislocalized to cytoplasmic foci. Proper localization of *hTR* to Cajal bodies can however be recovered by EXOSC10 depletion in PARN- or DKC1-depleted cells, suggesting that this is not the PARN depletion *per se* that is responsible for the mislocalization, but the destabilization of *hTR*.

We have added the following sentences in the introduction (page 4) to better explain the impact of PARN depletion on *hTR* localization in the cells: "Moreover, in the absence of PARN, the residual *hTR* was mislocalized into cytoplasmic foci. As exosome inactivation rescued *hTR* localization into Cajal bodies of PARN-depleted cells, it was suggested that PARN is not directly involved in *hTR*

localization into Cajal bodies but that the mislocalization results from an increased instability of hTR RNA in these cells (Shukla et al, 2016).”

Referee #2 (Comments on Novelty/Model System for Author):

The technical quality of this work is strong based on the visually apparent quality of gels, magnitude of biological effects (both absolute and relative), and the extent of quantitation and statistical tests. The novelty of this work is good because the study of PARN is still a relatively new topic in the telomere field, and the progress made by the authors that is reported in this manuscript is substantial. The medical impact is important given this is major mechanistic progress on the study of "telomeropathies" that are rare but often severe. Such diseases and disease mechanisms are an emerging field that spans multiple clinical fields.

The model system is patient samples and the mouse, and the results are clearly synergistic and complementary.

We thank the reviewer for his/her enthusiastic appreciation of our manuscript.

Referee #2 (Remarks for Author):

This work describes substantial new genetic insights into the role of PARN in the severe human telomeropathy, Høyerhaal-Hreidarsson (HH) syndrome. The study is important given the pinpointing of the specific gene mutations causing diseases in two patients, who were evaluated extensively. The data quality is very high, and the experiments are well-conceived and executed. The work lays the foundation for future study to understand how PARN operates and underlies this disease in molecular-mechanistic terms (it does not advance the study of PARN mechanism substantially *per se*). Overall, this is a very solid, innovative research study, about which I became increasingly enthusiastic as I read the authors' manuscript.

We thank the reviewer for his/her enthusiastic appreciation of our manuscript and suggestions. We provide point-by-point responses below.

That said, there is room for substantial improvement without extensive additional work. Foremost, the authors need to make it clearer what they believe their data add, if anything, regarding the mechanistic role of PARN in regulating hTR ncRNA and Shelterin-component mRNAs throughout the text (particularly by introducing this more extensively and then revisiting it in the Discussion and Abstract). If the authors do not conclude much about the mechanism(s) based on their data, then they alternatively need to explain in the Abstract and Discussion clearly that such questions as to how PARN is operating in regulating these RNAs remains largely unknown. As it stands presently, the key question of mechanism as to how PARN regulates these transcripts is left under-addressed, and this leaves the story, as written, more descriptive and less impactful than it could be if the authors commented further on this question as to PARN's functional activities.

Regardless of the degree to which the authors believe their data inform understanding of PARN mechanism, the Introduction also needs to inform the reader as to the current status of this question, beyond what is essentially just a single sentence (line 78 of page 3). Additionally, the authors should be clearer on line 83 of p. 4 as to what "modification" by PARN is being referred to: is it RNA nucleolytic cleavage or might this refer to some alternatively possible/known PARN enzymatic action beyond cleaving RNA?

→ We agree with the reviewer that our results do not provide *per se* mechanistic insights into the role of PARN in hTR and telomere-related gene regulation. We are confident however that the novel findings of our study are significant enough to justify a publication in *EMBO Molecular Medicine*. They can be summarized as follows:

- description of three **novel** germline *PARN* mutations causing Høyerhaal-Hreidarsson syndrome
- exhaustive clinical description of two unrelated patients
- demonstration that telomeres in fibroblasts from patients exhibit not only reduced telomere length and deprotection (TIF), but also, **for the first time**, increased instability
- **first** demonstration that short-term complementation of PARN KO human cells rescues telomere instability independently of any restoration of telomere length
- **first** demonstration that PARN depletion reduces the steady-state mRNA levels of human *TRF1*, *TRF2*, *TPP1*, *RAP1* and *POT1* shelterin genes independently of p53

- **first** demonstration that PARN depletion reduces the steady-state mRNA levels of human *DKC1* in a p53-dependent manner
- **first** demonstration that PARN does not regulate human TERRA levels
- **first** demonstration that PARN deficiency in HH patient-derived cells impairs rRNA biogenesis
- **first** demonstration that Parn haploinsufficiency in MEFs impairs rRNA biogenesis
- **first** demonstration that Parn KO mice are embryonic lethal at a very early stage and that the lethality is not reverted by p53 KO

On the other hand, we fully agree that, in its initial form, our manuscript was not helping the reader to appreciate the novelty of the findings and this is why, as the reviewer suggested, we considerably expanded the introduction to clarify the state-of-the-art in the field and the remaining open questions. We, as well, made corresponding changes in the abstract and the discussion to account for this remark.

As suggested by the reviewer we have also completed the introduction to detail the known activities exerted by PARN. We invite the reviewer to read the revised manuscript in order to appreciate the numerous changes that we made.

Related to mechanism and functional relationships between PARN and its telomere-affecting targets, Figure 8 could be improved. Is it really necessary to show the same model twice; one for functional PARN and one for dysfunctional? I would think this could be illustrated more cogently, and perhaps (related to the above paragraphs) convey some information as to what specific mechanistic action by PARN is required for each of its promoting and inhibiting effects on the listed targets.

→ According to the reviewer's suggestion, we modified Figure 8 by showing the model only once and by specifying, for each target, how PARN acts. We believe that this revised figure is much better and self-explanatory.

Finally, the authors do not really expand upon how novel the "pleiotropy" that they claim in the title, abstract, Discussion, etc. is for the field, nor precisely how they define it with respect to its targets and positive and negative roles that PARN has on them. The manuscript would be improved by the authors further expanding on why they decided this title was the most compelling by explaining if/why this is a new finding, and precisely in what respects.

→ We used the term "pleiotropic" in the title of our manuscript to highlight that PARN deficiency does not only cause telomere length defect, but also telomere instability and impaired rRNA biogenesis. However, it appears that the notion of "pleiotropic roles" is misleading and perhaps not well adapted. We therefore modified the title as follows: "Impaired telomere integrity and rRNA biogenesis in PARN-deficient patients and knock-out models". We have, as well, changed the text at various places to take this comment into account, mostly by referring to the pleiotropic consequences of PARN dysfunction, instead of the pleiotropic roles of the protein.

Referee #3 (Remarks for Author):

The authors have identified two unrelated Hoyeraal-Hreidarsson individuals carrying novel biallelic PARN mutations. Using PARN-mutated cells from patients and a human PARN KO cell line with inducible PARN complementation, Benyelles and coworkers have found that PARN deficiency affects both the length and the stability of the telomere, reduces the mRNA levels of a subset of shelterins and of *DKC1* and present aberrant ribosomal RNA biogenesis. The authors also generated a KO mice, which is embryonic lethal. The heterozygous mice also exhibit aberrant ribosomal RNA biogenesis.

Main concern

Although the different models (from patients, cells and mice) used in this work show robust and well-correlated results, the authors' findings are just the confirmation of previous published works. Thus, PARN deficiency in patients was reported to be accompanied by telomere defects several times already (Burris et al., 2016; Dhanraj et al., 2015; Moon et al., 2015; Tummala et al., 2015). Transient depletion of human PARN was also reported to be associated with a down-regulation of *DKC1*, *RTEL1* and *TRF1* transcripts (Tummala et al., 2015) and, similarly, experiments in mouse myoblast reported a decrease abundance of *Terf1*, *Terf2* and *Rtel1* gene transcripts (Lee et al. 2012).

Moreover, PARN-depleted cell lines present aberrant ribosomal RNA biogenesis (Ishikawa et al., 2017; Montellese et al., 2017). Because of all the above, I think the work from Benyelles and coworkers lacks of enough novelty to be published in *EMBO Mol. Med.*

→ We thank the reviewer for his/her positive appreciation of the originality of our models and the robustness of our results.

Although we do regret to read that the reviewer feels that our findings "lack enough novelty to be published in *EMBO Mol. Med.*", we acknowledge that the description of the state-of-the-art was not detailed enough to appreciate the novelty of our findings. We have extensively worked on the Introduction to account for this and we invite the reviewer to read it.

We would like to stress that, to the best of our knowledge, and by using unique cellular models (primary and SV40-transformed PARN-deficient cells from patient, human KO cell line complemented by wtPARN, Parn+/- MEFs), our study reports on several original results that clarify the multiple consequences of PARN deficiency. These hitherto undescribed results, that are of importance for a better understanding of the aetiology of this disease and of the multiple function of PARN, are listed below:

1. Three novel germline *PARN* mutations causing Høyeaal-Hreidarsson syndrome are reported in our manuscript, including p.Q68H (**Figure 1**) affecting an amino-acid extremely conserved across species.
2. We provide an exhaustive clinical description of two unrelated PARN patients with a detailed analysis of the immunologic features (**Table 2**). In particular, our immunological analysis demonstrates in both patients a reduction of circulating naive CD4+ T lymphocytes (CD31+ CD45RA+/CD4+) as well as a decreased in B (CD19+) and NK (CD16+CD56+) cells. To our knowledge, such detailed immunologic features, reporting the peculiar immunodeficiency often observed in HH patients, have not been reported in the previous articles describing PARN-deficient patients.
3. To the best of our knowledge, this is the first time that the analysis of telomere length and stability is performed in primary and transformed fibroblasts from PARN-deficient patients. This approach reveals for the first time that PARN-deficient cells from patients not only exhibit telomere length defect, TIF and senescence but also telomere instability (**Figure 2**).
4. Additionally, by using a human KO cell line complemented or not with wtPARN, we discovered that telomere instability caused by PARN deficiency is independent on telomere length (**Figure 4**). This is an important finding for the understanding of the aetiology of this disease.
5. Using an original set of cell lines (PARN KO cell line +/- complemented), we showed that PARN deficiency leads to a decreased abundance of several telomere-related gene transcripts (**Figure 3H**). This had never been shown before in human cells as Tummala et al. only reported on the impact of transient PARN depletion on the mRNA half-life of telomere-related genes in immortalized human cells, not on the steady-state levels of the transcripts (we have now detailed this in the Introduction). As unexpected results were obtained upon PARN knock-down in mouse myoblasts, where a decrease in transcript abundance could be associated with an increased stability of the affected mRNAs (Lee et al., 2012), the question remained open as to whether PARN defects down-regulate the steady-state levels of human telomere-related gene transcripts. Our findings that PARN KO is indeed associated with a down-regulation of several telomere-related gene transcripts is in agreement with the qRT-PCR results of Tummala et al. showing gene expression in "blood cells" from ctl and HH patients. Importantly enough however, one cannot be sure that these qRT-PCR results were not biased by the differences in blood cell populations. We believe that our model, using a human KO cell line complemented or not with PARN, provides more reliable results. We also showed that the impact of PARN depletion on *TRF1*, *TRF2*, *TPPI*, *RAP1* and *POT1* shelterin gene transcripts was independent on p53.
6. We, on the other hand, showed that the impact of PARN depletion on *DKC1* mRNA levels was dependent on the up-regulation of p53 in PARN KO cells. Again, this had never been showed before.
7. We showed that PARN deficiency does not alter TERRA levels (**Figure 3K**). Since TERRA represents a group of non-coding RNA with crucial role in telomere regulation, this observation provides an important information that had never been addressed before.
8. We (Montallese et al., 2017) and others (Ishikawa et al., 2017) recently reported on the role of PARN in rRNA biogenesis. However, in both articles the experiments were performed

in HeLa cells after PARN depletion with siRNAs. This approach, although informative, is not physiologic and does not prove that pathogenic *PARN* mutations causing HH provoke similar effects on rRNA biogenesis in the patient's cells. For the first time, we provide evidence, in PARN-deficient cells from two unrelated patients, that the rRNA biogenesis is altered (**Figure 6**). These observations were also supported by similar results obtained in PARN KO cell line +/- complemented and Parn+/- MEFs. It is noteworthy that our observation obtained in Parn+/- MEFs provides two supplementary original observations: (i) the role of PARN in rRNA biogenesis is conserved in mouse, (ii) Parn haploinsufficiency (at least in MEFs) impairs rRNA biogenesis (**Figure 7**).

9. We also report the first Parn KO murine animals that are embryonic lethal at a very early stage (**Figure 7**). This observation further supports the notion that the clinical features observed in PARN-deficient patients not only relies on telomere maintenance defect as initially thought (see Discussion).
10. Lastly, we demonstrated that the lethality of Parn KO animals was not reverted by p53 KO (**Figure 7**). Provided that the up-regulation of p53 mRNA and/or protein has been proposed by different studies to be an important contributor to the phenotype, this observation was important and fits with our discovery that the impact of PARN KO on most telomere-related gene transcripts or rRNA biogenesis occurs independently of p53.

2nd Editorial Decision

10 April 2019

Thank you for the submission of your revised manuscript to EMBO Molecular Medicine. We have now received the enclosed report from the referee who was asked to re-assess it. As you will see this reviewer is now globally supportive and I am pleased to inform you that we will be able to accept your manuscript pending minor editorial amendments and the text changes requested by Referee #1.

***** Reviewer's comments *****

Referee #1 (Remarks for Author):

The manuscript has been extensively revised and improved. My comments have been addressed, except for the diagnosis of aplastic anemia, which is still confusing and clinically inaccurate. Line 138: bone marrow aspirate does not show aplastic anemia, as it is not diagnostic. A bone marrow biopsy is required for diagnosis (along with blood counts etc.). I would strongly recommend the authors to revise the diagnosis of patients with the hematologists to make sure the clinical information is accurate.

2nd Revision - authors' response

24 April 2019

Authors made the requested changes.

Corresponding Author Name: Patrick Revy & Anabelle Decottignies

Manuscript Number: EMM-2018-10201V2